# Agreement-Based Cascading for Efficient Inference

**Steven Kolawole**\*                                        *skolawol@andrew.cmu.edu*
*Carnegie Mellon University*

**Don Dennis**\*                                             *dondennis@cmu.edu*
*Carnegie Mellon University*

**Ameet Talwalkar**                                          *talwalkar@cmu.edu*
*Carnegie Mellon University*

**Virginia Smith**                                           *smithv@cmu.edu*
*Carnegie Mellon University*

**Reviewed on OpenReview:** *https: // openreview. net/ forum? id= jn9B7LMlzk*

## Abstract

Adaptive inference schemes reduce the cost of machine learning inference by assigning smaller models to easier examples, attempting to avoid invocation of larger models when possible. In this work we explore a simple, effective adaptive inference technique we term *Agreement-Based Cascading (ABC)*. `ABC` builds a cascade of models of increasing size/complexity and uses agreement between ensembles of models at each level of the cascade as a basis for data-dependent routing. Although ensemble execution introduces additional expense, we show that these costs can be easily offset in practice due to large expected differences in model sizes, parallel inference execution capabilities, and accuracy benefits of ensembling. We examine `ABC` theoretically and empirically in terms of these parameters, showing that the approach can reliably act as a drop-in replacement for existing models and surpass the best single model it aims to replace in terms of both efficiency and accuracy. Additionally, we explore the performance of `ABC` relative to existing cascading methods in three common scenarios: (1) edge-to-cloud inference, where `ABC` reduces communication costs by up to 14×; (2) cloud-based model serving, where it achieves a 3× reduction in rental costs; and (3) inference via model API services, where `ABC` achieves a 2-25× reduction in average price per token/request relative to state-of-the-art LLM cascades.

## 1 Introduction

The high cost of inference associated with deploying large machine learning (ML) models presents a significant barrier to their adoption (Strubell et al., 2020; Kaplan et al., 2020). As models continue to increase in size, practitioners are often faced with investing substantial resources in updating existing deployments or settling for lower-performing alternatives. However, for many applications, it has been shown that a considerable portion of the data seen during inference can be effectively evaluated using small models rather than large, state-of-the-art models (Chen et al., 2020; Jitkrittum et al., 2023). This means that if we can identify the subset of data samples that can be accurately evaluated by more inexpensive models, average inference costs can be reduced. This problem is often referred to as *adaptive inference*, where the cost of inference adapts to some notion of 'difficulty' of each example seen at inference time.

A natural approach for adaptive inference is to *cascade* over a set of potential models, starting from the least expensive and moving to more expensive models based on some *deferral rule* (Rowley et al., 1998; Viola & Jones, 2001; Soo, 2014). A common cascade construction is to use a Pareto-efficient set of models and an

---

\*Equal contribution.

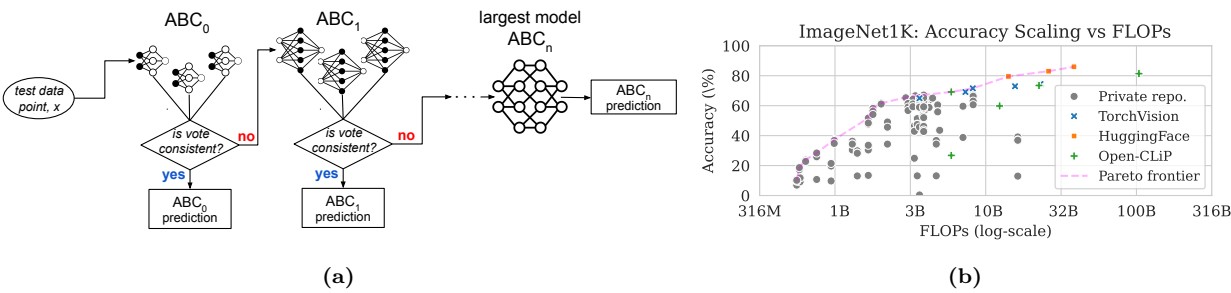

**Figure 1: (a)** Agreement-Based Cascading (`ABC`): `ABC` introduces a data-dependent routing scheme that uses agreement amongst an ensemble of models to determine whether to cascade to larger models. If the predictions of smaller ensembles do not align, the cascade moves to the next tier of larger models, continuing until agreement is reached or the largest model(s) are used. This can reduce cost by limiting the use of the largest models to cases where smaller models cannot reach consensus. **(b)** `ABC` is a natural baseline for adaptive inference due to (i) the vast number of pretrained models available to ML practitioners today; (ii) the fact that even small accuracy gains often require an order-of-magnitude increase in FLOPs, mirroring proposed scaling laws and resulting in large differences in model sizes between cascade tiers (Hestness et al., 2017; Henighan et al., 2020; Madaan et al., 2023). The pink dashed line represents the Pareto-optimal frontier, showing the models with the highest accuracy for a given computational budget. We show that `ABC` can effectively improve this frontier—allowing practitioners to achieve high accuracy without incurring the full computational cost of the largest models by invoking smaller models for 'easier' samples.

easy-to-compute deferral rule such as the confidence scores of the models predictions (Viola & Jones, 2004; Wang et al., 2018a; 2021). Recently, cascading has seen renewed interest in the ML community with the advent of large language models (LLMs) and vision foundation models (e.g., Chen et al., 2023; Gupta et al., 2024). Recent approaches consider a variety of techniques to construct model cascades, including designing novel model architectures that have built-in cascading capabilities (Cai et al., 2019; Devvrit et al., 2023; Khare et al., 2023), or learning routing schemes/deferral rules that require data-dependent training for every task considered (Chen et al., 2023; Ding et al., 2024). While these approaches can improve accuracy/efficiency trade-offs, they may also introduce significant computational overhead in setup and training/finetuning costs.

In this work, we instead investigate a simple, *training-free* cascade scheme that uses the agreement among an ensemble of existing models at each cascade level as its deferral rule. We refer to this approach as *Agreement-Based Cascading (`ABC`)*. Intuitively, when the outputs of the ensemble at a particular cascade-level do not align, `ABC` triggers cascading, and inference is attempted at the next tier of (larger) models (see Figure 1). This scheme has been explored historically for specialized applications such as face detection (Zuo & de With, 2005; 2008; Susnjak et al., 2012), where ensembles of binary face detectors focus on subregions of the face and combine their output using decision networks. However, to the best of our knowledge, our work is the first to study the use of ensemble agreement as a deferral mechanism for more general, modern machine learning workloads.

We point to a few trends in ML that make `ABC` a particularly attractive approach for model cascading. In particular, while using an ensemble of models at each level may initially appear to increase overall inference costs, it is natural to believe this simple baseline could excel in real-world applications due to: (1) the growing ease of obtaining pretrained models of various sizes and accuracies due to the rise of model registries (Wolf et al., 2019) as well as flexible compression schemes (Zhu et al., 2023; Dennis et al., 2023); (2) scaling laws that predict a large increase in inference cost for every marginal increase in accuracy (Hestness et al., 2017) (Figure 1); and (3) an increased ability to execute ensembles of relatively small models in parallel, with minimal additional costs (Fern & Givan, 2003; Kim et al., 2023; Miao et al., 2023) (Figure 3).

With these motivations in mind, we rigorously study `ABC` as a baseline for adaptive inference in modern ML workloads. Overall, we make the following contributions:

- We propose `ABC` as a training-free, deferral-based cascading approach that leverages agreement among ensembles of existing models, removing the need for additional routing networks or specialized architectures.

- We theoretically characterize cases where `ABC` can replace an existing model deployment without affecting the accuracy, and define safe deferral rules as sufficiency conditions for their existence. We further characterize the expected accuracy and inference costs in this setting.

- We empirically evaluate `ABC` on a wide range of image and language tasks and find that `ABC` not only improves efficiency, but also accuracy, compared to the model that it aims to replace. We then consider

the performance of `ABC` relative to existing cascading methods in common inference scenarios, including (1) edge-to-cloud inference where `ABC` reduces communication costs by up to 14×, (2) model-serving on heterogeneous GPUs, where `ABC` reduces rental costs by up to 3× and (3) inference using black-box access to model API services, where `ABC` shows up to a 25× reduction in average price per token.

## 2 Related Work

Adaptive inference schemes have been a topic of interest in machine learning for many years. This section discusses three main approaches to cascading and adaptive inference: score-based methods, trained routers, and dynamic networks. We highlight how `ABC`'s approach relates to and differs from these existing methods.

### 2.1 Cascading using Score-based Deferrals

Traditional cascading methods often rely on simple *score-based metrics* for deferral decisions. We compare to the recent Wisdom-of-Committees method of Wang et al. (2021) as a general, representative method in this category but note that specialized instantiations of this approach have been applied across various domains in prior work, for instance, in object detection (Rowley et al., 1998; Viola & Jones, 2004; Wang et al., 2011; Cai et al., 2015; Angelova et al., 2015; Streeter, 2018), image classification (Wang et al., 2018a; 2021), and text classification (Li et al., 2021b; Mamou et al., 2022; Varshney & Baral, 2022; Lebovitz et al., 2023). In these settings, deferrals within cascading systems use metrics such as confidence scores, entropy, and probabilities (Gangrade et al., 2021; Geifman & El-Yaniv, 2019; Narasimhan et al., 2022). While using confidence scores of existing models is inexpensive, these scores are required to be well-calibrated, which is less common for off-the-shelf models (Guo et al., 2017; Enomoro & Eda, 2021). Recent methods have thus considered a variety of techniques to enhance score-based deferrals, such as explicitly adding deferrals as a prediction (Wang et al., 2018a), difficulty-aware regularization (Li et al., 2021b), temperature scaling (Wang et al., 2023), and calibration mechanisms (Nie et al., 2024).

Recently, Jitkrittum et al. (2023) explored the specific practical settings under which confidence-based deferral in cascades could suffer; these settings include: scenarios where downstream models are specialists (the error probability of the downstream model is highly non-uniform across samples), when samples are subject to label noise, and in the presence of a distribution shift between the train and test set. Notably, in all these failure modes, where confidence-based cascades tend to be inadequate, our approach is likely to offer improvements, as ensembles are known to help induce diversity, enable robustness to noise, and mitigate issues of distribution shift (Gontijo-Lopes et al., 2022; Sharkey, 1996; Dietterich, 2000; Džeroski & Ženko, 2004).

### 2.2 Cascades with Routing Procedures

Other methods avoid confidence scores entirely by training procedures independent of the model set to route instances as needed. This trend has gained prominence with the rise of black-box model API services for LLMs, where prediction scores are often unavailable. Guan et al. (2018) developed a selection module trained to determine the best-fit classifiers for data instances. Yue et al. (2024) (MoT LLM Cascade) used sampling and consistency checking to determine when to defer to a high-cost model. AutoMix (Madaan et al., 2023) proposed a few-shot self-verification mechanism, similar to Yue et al. (2024), but also introduced a Markov-based meta-verifier for cascading in context-grounded tasks. FrugalGPT (Chen et al., 2023) leveraged a cascade strategy that triages incoming queries using a DistilBERT router and scoring function. HybridLLM (Ding et al., 2024) used a fine-tuned DeBERTa (He et al., 2020) to route queries to models based on the predicted query difficulty and the desired quality level. On the same note, OrchestraLLM (Lee et al., 2023)—using hand-labeled data to create expert model pools—selects model to query based on embedding distances between the pools' instances and the test instances. Other methods also used some form of trained router, including RouteLLM (Ong et al., 2024), 'Fly-swat or cannon' (Šakota et al., 2024), and Shnitzer et al. (2023).

As we further discuss in Section 5.2.3 (where we empirically compare to FrugalGPT, AutoMix, and MoT LLM Cascade), these more sophisticated methods involve complex setups, data-dependent training, and increased computational overhead. They often require retraining routers for each new task, dataset, or model,

limiting adaptability to unseen data distributions or new models without incurring further costs. In contrast, `ABC` provides a simpler, flexible, widely applicable alternative at no additional training or setup cost.

### 2.3 Dynamic and Adaptive Networks

Routing procedures typically train an independent routing mechanism, keeping the Pareto-set of models untouched. However, for many medium-scale applications, methods have been explored that learn a Pareto-set of models and stopping criterion together. For example, early exit methods of Bolukbasi et al. (2017); Huang et al. (2018); Shafiee et al. (2018); Wang et al. (2018b); Hu et al. (2020); Xin et al. (2020); Geng et al. (2021); Zhou et al. (2020); Liu et al. (2020); Schuster et al. (2022), subnetworks extraction methods (Yu et al., 2018; Yu & Huang, 2019; Chen et al., 2021; Hou et al., 2020; Han et al., 2022; Devvrit et al., 2023), composition-based methods (Suggala et al., 2020; Dennis et al., 2023; Du & Kaelbling, 2024) are some of the more recent examples. While these methods achieve impressive efficiency gains, they require training specialized architectures from scratch with adaptive capabilities.

Our approach diverges from these methods, as it does not involve altering model architectures or retraining from scratch. Instead, `ABC` leverages existing models as they are, utilizing ensemble agreement as a deferral condition, which can be applied directly to these models without any need for fine-tuning or specialized training.

## 3 Agreement-Based Cascading (`ABC`)

In this section, we present a high-level overview of the Agreement-Based Cascading (`ABC`) approach, providing the essential concepts and insights needed to understand `ABC`'s effectiveness in the experiments (§5). Readers interested in the formalizations and technical details can refer to §4 for a more comprehensive treatment.

### 3.1 Overview of `ABC` Approach

As outlined in Section 1, the goal of adaptive inference is to identify data samples that can be evaluated accurately by a relatively inexpensive model. This way, we can reduce the inference cost based on whether a sample is 'easy' or 'hard'. A common approach to this problem is deferral-based cascading, where we cascade over a sequence of models, starting from the least expensive and using a 'deferral rule' to determine if a higher tier model must be used. These deferral rules are typically significantly cheaper to evaluate (e.g. a small neural network), and add very little additional cost on top of model execution. This approach makes it so that 'simpler' cases are managed by smaller, faster models, while only the complex cases cascade up multiple tiers and to more resource-intensive models.

*Agreement-Based Cascading (ABC)*, described in Algorithm 1, is one such deferral based cascading approach. `ABC` maintains a set of ensembles $\{H_1, \ldots, H_{n_E}\}$ that starts from an ensemble of inexpensive models in $H_1$ to expensive, state-of-the-art models in $H_{n_E}$. Similar to other cascading techniques, given a sample $x$, we start inference from the lowest (cheapest) tier in the cascade and use a deferral rule, $r_i(x)$, to determine if a higher tier model is needed. A core distinction between `ABC` and existing cascading approaches is its agreement-based deferral rule. Instead of training a small additional deferral network or post-hoc deferral rules, we use a notion of *agreement* between models within each ensemble as a confidence measure. When a (configurable) fraction of models in a tier agree, it signals that they likely have the right answer. Conversely, if they disagree, the uncertainty triggers a deferral to the next, more powerful ensemble. Additionally, we focus on cases where we wish to maintain accuracy of the overall prediction and only improve inference cost when accuracy does not suffer. This is different from existing approaches that prioritize a fixed inference budget, potentially at the cost of accuracy.

### 3.2 `ABC` in Modern ML Environments

In recent years, obtaining trained models has become much easier, thanks to the rise of public and private model repositories (Wolf et al., 2019), where a wide variety of pre-trained models are readily available. This abundance of accessible models of various sizes and performance levels makes the practical application of

---

**Algorithm 1** Agreement-Based Cascading (`ABC`)

---

**Require:** Set of ensembles $\{H_1, H_2, \ldots, H_{n_E}\}$, deferral rule $r_i$ for each ensemble $i \in [n_E]$ as in Equation 3 or 4

**Require:** A new inference data point $x$.

1: Current cascade level, $i \leftarrow 1$
2: Cascaded prediction, $y \leftarrow \emptyset$
3: **for** $i \in \{1, \ldots, n_E\}$ **do**
4:     $y \leftarrow H_i(x)$
5:     **if** $r_i(x) = 0$ **then**
6:         **break**                                              {Models in ensemble 'agree'}
7:     **end if**
8: **end for**
9: **return** $y$

---

`ABC` especially appealing. Unlike methods that require additional task-specific training or fine-tuning, `ABC` can leverage these existing models, making it suitable for direct deployment as a "drop-in" replacement for high-cost models. This adaptability, combined with minimal setup and training requirements, can allow `ABC` to fit seamlessly into many existing ML workflows.

Strictly speaking, evaluating agreement between multiple models in an ensemble is expensive when compared to the small router models that are used in many existing approaches. However, two key aspects of modern ML workloads can help to mitigate this additional cost in practice. First, in many cases some degree of *parallelization* is available that can reduce the impact on inference cost metrics such as latency. We use $\rho$ to smoothly interpolate between the fully sequential case ($\rho = 0$) to the fully parallel case, ($\rho = 1$) as detailed in §4.3. Second, in many use-cases the difference in cost between models in successive tiers of cascades is so large that the impact of lower tier models on the overall cost of inference is negligible, even with the added cost of constructing ensembles (see Figure 3). We use $\gamma$ to denote the *relative cost* of the models — the ratio of the cost of the smaller model to the larger model.

In §5, we evaluate `ABC`'s performance across several real-world scenarios (resulting in varying settings of $\rho$ and $\gamma$), demonstrating its competitiveness and efficiency. We first show that `ABC`'s ensemble-based deferral mechanism preserves and often improves on the accuracy of models it aims to replace. We also show that when the lower tier models are small enough compared to the higher tier models, `ABC`'s ensemble-based deferral rule operates with negligible impact on the overall cost.

We then evaluate cases in real-world workloads where such large relative costs between various tiers of `ABC` is natural. In edge-to-cloud setups, `ABC` enables substantial reductions in communication costs by processing simple tasks locally on the edge. This minimizes the need to transfer data to the cloud, reducing both latency and data transfer expenses. When serving models on heterogeneous GPU resources in the cloud, `ABC` significantly reduces inference costs. By selecting models based on task complexity, `ABC` makes efficient use of available GPU resources, optimizing for both accuracy and rental cost. In black-box API-based model deployments, where the user is billed per request or token, `ABC` offers substantial savings by reducing the average cost per request. By deferring to high-cost models only when necessary, `ABC` achieves notable economic savings compared to standard inference approaches.

## 4 Formalization and Theoretical Details

In this section, we formalize our problem setup by first outlining the standard statistical learning framework, which serves to define the concepts of models, ensembles, the learning performance (e.g., accuracy) and the inference cost associated with ensembles (§4.1). Next, we describe deferral rule-based cascades and formulate the specific case of *drop-in cascades*, cascades which prioritise accuracy over inference cost savings (§4.2). We introduce *safe deferral rules* as a sufficiency condition for constructing drop-in cascades and define our deferral rule based on agreement between models in an ensemble (§4.3). We conclude this section by formalizing the test-time accuracy-inference cost behaviour of drop-in cascades (§4.4).

### 4.1 Problem Setup and Notation

Consider the standard statistical learning settting where $\mathcal{X}$ denotes an instance space and $\mathcal{Y}$ denotes the label or response space. Let $h(x)$ be a model taking inputs from $\mathcal{X}$ and producing outputs in $\mathcal{Y}$. We assume that the models are from some hypothesis class $\mathcal{H}$. In this setup, we typically characterize the learning performance of various models using its *risk* with respect to some data distribution $\mathcal{D}$ over $\mathcal{X} \times \mathcal{Y}$ and a loss function $l$, given by,

$$\mathcal{R}_l(h) = \mathbb{E}_{(x,y)\sim\mathcal{D}}[l(h(x), y)].$$

As a concrete example, in the classification setup, where we use the mis-classification error as a loss function, the risk is given by $\mathcal{R}(h) = \mathbb{E}(y \neq h(x))$.

Assume that each model $h \in \mathcal{H}$ has a cost of inference, denoted by $C : \mathcal{H} \to \mathbb{R}_+$; for instance, for cases where we are concerned about inference latency, the cost can be the latency of the model on the target hardware. Let $h_1$ and $h_2$ be two models with $h_2$ being the more expensive of the two. We denote the *relative cost* of the models by $\gamma := \frac{C(h_1)}{C(h_2)}$, satisfying $0 < \gamma \leq 1$.

Let $H^k : \mathcal{X} \to \mathcal{Y}$ denote an ensemble of $k$ models from the same hypothesis class $\mathcal{H}$. Similarly to before, the learning performance of various ensembles can be characterized by their risk; let $\mathcal{R}(H^k)$ denote the risk of an ensemble $H^k$. Compared to a single member model, the cost of evaluating the entire ensemble of models depends on the various factors, including the degree of parallelization. Assuming that the models in an ensemble are of similar cost, say $c_0$, we model the cost of the ensemble using a *parallelism coefficient* $0 \leq \rho \leq 1$ as,

$$C(H^k) = c_0 k^{1-\rho}. \tag{1}$$

Here, when $\rho = 1$, the ensemble suffers the same cost as a single model and corresponds to the case where models can be fully parallelized. On the other extreme, at $\rho = 0$, the cost of the ensemble of $k$ models is $kc_0$, corresponding to no parallelization (sequential evaluation).

### 4.2 Deferral Rule-Based Drop-in Cascade

A deferral-based cascade consists of a finite set of models and a deferral rule. Here, the idea is to start with the most resource-efficient model and use the deferral rule to determine if a better-performing model with a higher resource cost should be used for the current sample. The deferral rules themselves are designed to have negligible inference cost given the models inference outputs. For this exposition, we will restrict ourselves to cascades with only two levels, though the discussion in this section can be readily generalized to larger cascades. Let the cascade consist of an ensemble $H_1^k$ at the lower level, and the larger model $h_2$ at the higher level; $\mathcal{M} = \{H_1^k, h_2\}$. Let $r(x)$ denote a deferral rule;

$$r(x) = \begin{cases} 1 & \text{defer to } h_2 \\ 0 & \text{use } H_1^k. \end{cases}$$

In this work, we focus on deployment scenarios where a decrease in model performance is a critical concern. This is particularly relevant when cascades are intended to function as a drop-in replacement for an existing deployment of an expensive model. This notion can be formulated as maximizing the number of calls to the smaller model while retaining the accuracy of the larger model. With a slight abuse of notation, let $\mathcal{M}_r(x)$ denote the prediction of the cascade with $\mathcal{M} = \{H_1^k, h_2\}$ and deferral rule $r$. Then we desire that for a choice of a small error budget $\xi > 0$,

$$\max_r \quad \mathbb{P}(r(x) = 0)$$
$$\text{s.t.} \quad \mathbb{P}(y \neq \mathcal{M}_r(x)) \leq \mathbb{P}(y \neq h_2(x)) + \xi, \tag{2}$$

The constant function $r(x) = 1$, which defers for every $x$, is feasible and attains the objective value of 0. Moreover, *every* feasible deferral rule leads to a cascade with competitive accuracy as $h_2$. Instead of picking

an error budget $\xi$, a complimentary (dual) approach is to consider a fixed inference budget and aim to attain the best accuracy within the budget. In this view, drops in accuracy (compared to the existing large model) are permissible, provided the inference budget is met. Jitkrittum et al. (2023) examine this perspective, which we recommend to interested readers.

### 4.3 Deferral using Ensemble Agreement

Ensemble agreement is an instantiation of score based deferral rules (Jitkrittum et al., 2023). In its simplest form, we associate a score to the prediction output of an ensemble model and interpret this score as a measure of confidence in that particular prediction. We can then defer to a larger model when the confidence is below a predetermined threshold. For a model $H_1^k(x)$ and a scoring function $s(x)$, the deferral rule is defined as

$$r(x) = \mathbb{I}[s(x) \leq \theta],$$

where $\mathbb{I}$ is the indicator function. In general, such score based deferral rules can sometimes be misleading. For instance, in multi-class classification, rules that map the outputs of a classifier to a probability distribution over labels can produce confidently incorrect predictions when an unperceivable amount of perturbation is added to the data sample (Szegedy, 2013). However, such rules have been shown to be effective in practice (Wang et al., 2018a; Gangrade et al., 2021; Mamou et al., 2022; Gupta et al., 2024) implying that such adversarial data is rare in many cases. Motivated by this observation, we propose the following property of scoring functions (and by extension the corresponding deferral rule).

**Definition 4.1** (Safe deferral rule). *Let $H^k : \mathcal{X} \to \mathcal{Y}$, $k \geq 1$ be a classifier and let $s : \mathcal{X} \to [0, 1]$ be a scoring function. The scoring function $s$ is referred to as a safe scoring function for $H^k$ if there exists $\theta \in [0, 1]$ such that, for some small $\epsilon > 0$,*

$$\mathbb{P}(s(x) \geq \theta, H^k(x) \neq y) \leq \epsilon.$$

*The corresponding deferral rule $r(x) = \mathbb{I}[s(x) \leq \theta]$, is referred to as a safe deferral rule, and satisfies $\mathbb{P}(r(x) = 0, H^k(x) \neq y) \leq \epsilon.$*

We define safe deferral with respect to classification tasks for simplicity. The definition can be extended to other cases with appropriate choice of loss function. Intuitively, this formalizes the notion that the deferral rule can probabilistically identify a *subset* of $\mathcal{X} \times \mathcal{Y}$ for which $H_1^k(x)$ is correct. Alternatively, we can think of the deferral rule as a one sided classifier, similar to the problem studied in Goyal et al. (2020), where the class to be identified is the *subset* of $\mathcal{X} \times \mathcal{Y}$ where $H_1^k(x)$ is correct. In general, such a safe deferral rule need not exist for a given model $H_1^k(x)$. However, in §5 we demonstrate empirically that such rules can in-fact be constructed without additional training for many real world tasks with appropriate choice of $\theta$ (Figure 7). These rules lead to selection rates as high as 90% of the data in cases such as ImageNet-1K.

Assuming access to a safe deferral rule, observe that for $\xi \geq \epsilon > 0$, such rules are feasible for the program in Equation 2 (see Proposition 4.1). This implies that *every* cascade, $\mathcal{M} = \{H_1^k, h_2\}$ using a safe deferral rule $r$, is competitive with the larger model $h_2$ in terms of accuracy. We empirically evaluate these drop-in cascades in §5. Note that the optimal safe deferral rule can lead to an improvement in the accuracy of ABC in theory (see, Appendix A), and we see accuracy improvements in our experiments (§5).

**Agreement-based deferral rule.** We evaluate two flavors of deferral rules that capture the notion of agreement between models. In cases where we have direct access to the models, we directly use the outputs produced by models in the ensemble. However, in certain cases, for instance when interfacing through a third-party provider's inference API, we only have black box access to models. For such cases, we use a voting scheme between the models in an ensemble to construct our scoring function. Concretely, for an ensemble $H^k$ consisting of $k \geq 1$ models, let $s(x; H^k)$ denote the average score of the majority prediction on $x$ and let $\text{vote}(x; H^k) = \frac{1}{k} \sum_{h \in H^k} \mathbb{I}[H^k(x) = h(x)]$ denote the fraction of votes received by the prediction, where,

$$r_\text{v}(x; \theta_v) = \begin{cases} 1 & \text{vote}(x; H_1^k) \leq \theta_v \\ 0 & \text{otherwise.} \end{cases} \tag{3}$$

$$r_\text{s}(x; \theta_s) = \begin{cases} 1 & s(x; H_1^k) \leq \theta_s \\ 0 & \text{otherwise.} \end{cases} \tag{4}$$

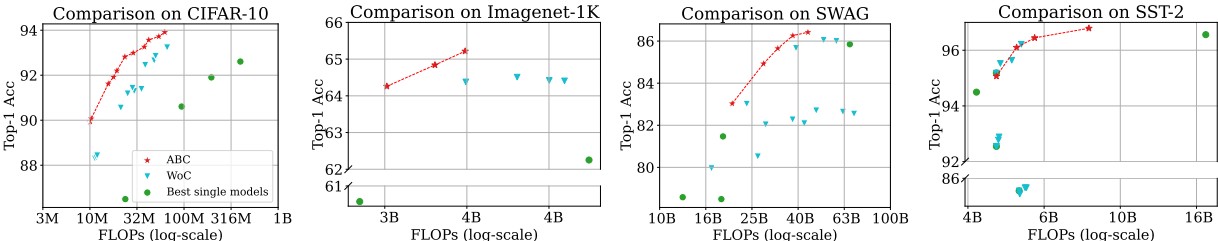

**Figure 2:** Pareto curves of `ABC` vs. confidence-based cascades (WoC) (Wang et al., 2021) and best single models on diverse tasks. For WoC, we tune its cascade configurations across the best four of its confidence thresholds and generate results from their most performant cascades. `ABC` maintains a Pareto-optimal curve, which consistently outperforms both methods in accuracy with lower FLOPs costs.

### 4.4 Inference Cost Savings and Competitiveness

Since we do not impose an inference cost budget on the cascade, it is possible for the cascade to incur a higher cost than simply using the larger model $h_2$. Intuitively, if majority of the samples seen during test time are 'hard', the cascade pays the cost of evaluating both $H_1^k(x)$ and $h_2(x)$.

**Proposition 4.1.** *Let $\mathcal{M} = \{H_1^k, h_2\}$ be two classifiers and $r$ a deferral rule such that $r$ is a safe deferral rule for $H_1^k$ according to Definition 4.1, for a distribution $\mathbb{P}$ over $\mathcal{X} \times \mathcal{Y}$. Then for every $\xi \geq \epsilon > 0$, the agreement based cascading (ABC) classifier $\mathcal{M}_r(x)$ is such that,*

*1. The ABC classifier is competitive with the large classifier $h_2$ in terms of accuracy (zero-one loss),*

$$\mathcal{R}(\mathcal{M}_r) \leq \mathcal{R}(h_2) + \epsilon,$$

*2. The ABC classifier enjoys an average inference cost of,*

$$\mathbb{E}[C(\mathcal{M}_r)] = (k^\rho \gamma + \mathbb{P}(r(x) = 1))C(h_2).$$

The proof follows directly from Definition 4.1 and basic probability (see Appendix A). Safe deferral rules can be constructed with minimal cost: our threshold estimation requires only ∼100 validation samples and simple voting computations, avoiding expensive router training. Thus, the cost savings we can expect with a drop-in cascade depend on three key factors: the relative cost $\gamma$, the degree of parallelization $\rho$ and the deferral rate $P(r(x) = 1)$, or equivalently, the *selection rate* $P(r(x) = 0)$. In particular, for *every* feasible deferral rule $r$, in the best case scenario where the cost of the smaller model is negligible (i.e., $\gamma = 0$), the cost of inference reduces by the selection rate, $P(r(x) = 0)$. Conversely, in the worst-case scenarios, the cost can be $(k + 1)$ times the cost of the larger model. In the next section, we demonstrate real world scenarios where the favorable interplay of these three quantities lead to significant improvements in inference cost. See Appendix C and Figure 7 for information on selection rates for various models and datasets considered here.

## 5 Experiments

This section evaluates our training-free Agreement-Based Cascading (`ABC`) approach across a variety of language and vision tasks, focusing on accuracy and inference cost. First, in §5.1.1, we examine `ABC`'s accuracy-cost tradeoff against state-of-the-art models under full parallelization, setting the *parallelization factor* ($\rho$) to $\rho = 1.0$ for optimal tradeoff. Next, in §5.1.2, we show that using ensembles at lower tiers has minimal impact on inference cost when lower-tier models are at least $50\times$ cheaper than the largest model. In such cases, the *relative cost* ($\gamma$) satisfies $\gamma \leq \frac{1}{50}$, making `ABC` effective without parallelization. Finally, §5.2 explores practical scenarios where low relative costs make `ABC` is naturally suited for real-world deployments.

**Estimating Voting Threshold:**   `ABC`'s deferral rule uses a configurable voting threshold, $\theta$ (see Equations 3 and 4) at each cascading tier. We estimate $\theta$ empirically on a small set of unseen data; see App. B for details.

**Datasets:** To evaluate `ABC`, we use a range of benchmark datasets for image and language tasks, as shown in Table 2 in the Appendix. Additional datasets are used in §5.2.3 to align with those explored by state-of-the-art baselines.

**Models:** We select diverse models for both image and language tasks, summarized in Appendix's Table 3. For BERT-based models, we use the `BASE` and `LARGE`, and for image models, we tier by FLOPs count. All models are sourced from HuggingFace Zoo for inference without any additional training effort on our end. §5.2.3 uses models from LLaMA 3, Gemma 2, and Qwen 2 families via the Together API. We detail this section's experimental setup in App. D.2.

**Evaluation:** For the generation tasks we consider in §5.2.3, the datasets each consist of a fixed set of possible outputs akin to the classification tasks, and we apply our deferral rule on the final output at each tier. Certain cases don't have a fixed set of output labels (while still not being open-ended generation), like (1) GSM8K, where there is a final numeric answer at the end of the generated output to the math word problems, and (2) CoQA, where we used F1-score to capture overlaps between predictions and ground-truth answers. We do not discuss open-ended generation tasks in this work as the deferral rule we consider—and the baseline methods that we compare to—is not directly applicable to such cases.

## 5.1 When is `ABC` Practical?

### 5.1.1 When Parallelization is Cheap

We first consider the case where the inference cost of an ensemble of models is the same as the cost of a single model. This idealistic scenario could happen, for instance, in offline batch inference (Jetty et al., 2021), when GPUs are available for parallelization, or the models are small enough that existing resources are under-utilized and an ensemble adds no additional cost. More importantly, this setting establishes best-case accuracy and inference cost values for `ABC`. We consider total floating-point operations (FLOPs) as a representative inference cost metric here, and discuss other metrics such as communication cost, latency, or cloud rental costs in subsequent subsections.

The accuracy vs. FLOPs for `ABC` is shown in Figure 2. As a point of comparison, we also include Wisdom-of-Committees (WoC) (Wang et al., 2021), a popular and representative confidence-based model cascading method. In most cases, we observe that `ABC` is able to improve the Pareto frontiers, as it usually sees a 1–2 point increase in accuracy. We attribute this to a combination of (a) the improvement ensembles can have on accuracy that is widely noted in literature (Gontijo-Lopes et al., 2022; Jiang et al., 2023) and (2) the improvement agreement-based rules can cause in cascading (Appendix A). Overall, in terms of accuracy, `ABC` either exceeds—or at least matches—the accuracy of the best models at a fixed FLOPs budget, and is a practical drop-in replacement for these models.

### 5.1.2 Disparity in Relative Cost is High

Of course, in many real-world scenarios, the cost of evaluating ensembles is not negligible. For instance, in the edge-to-cloud inference scenario discussed in §5.2.1, models in an ensemble often

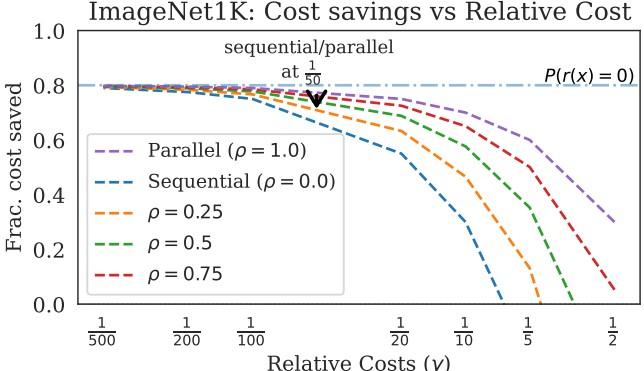

**Figure 3:** Fraction of inference cost saved as a function of relative cost of models ($\gamma$), *assuming* a fixed selection rate $\mathbb{P}(r(x) = 0)$. As parallelization decreases from fully parallel ($\rho = 1$) to sequential ($\rho = 0$), cost of evaluating ensembles increase and the cost savings decrease. When models across tiers are of similar size (e.g., smaller model is at most 5× smaller, $\gamma \geq \frac{1}{5}$), some parallelization is needed for `ABC` to reduce costs effectively. However, for lower relative costs (e.g., smaller model is at least 50× smaller, $\gamma \leq \frac{1}{50}$), sequential and parallel settings achieve meaningful savings, showing `ABC`'s efficiency even with the added cost of using ensembles.

are evaluated sequentially. However, even in such scenarios, if the relative cost of models across each level of the cascade, $\gamma$, is small enough, this additional cost becomes negligible compared to the overall cost.

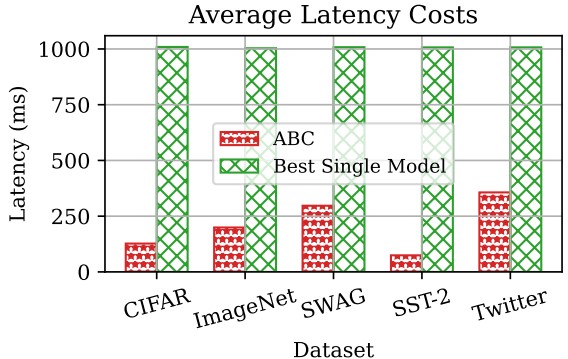 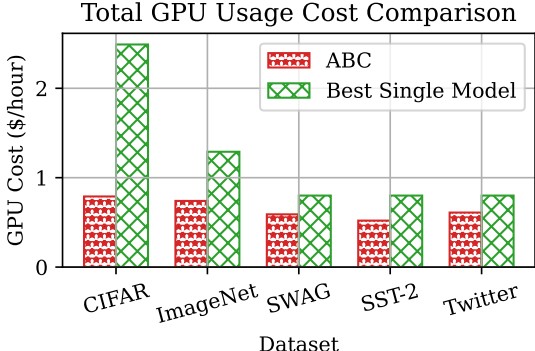

**Figure 4:** (a) `ABC` for edge-to-cloud inference: We simulate a single-instance inference setup, as seen in real-time applications where predictions may need to be made as new data becomes available. `ABC` can enable small models to be served at the edge without sacrificing accuracy—leading to large savings in communication costs over the alternative of using only the highest accuracy/largest model residing in the cloud, or a single small and low-performing model on the edge. (b) Total GPU usage costs of `ABC` vs. using the best model. Agreement-Based Cascading, at reduced costs of GPU usage, exceeds the accuracy of the single best models in all task categories.

In Figure 3, we demonstrate the impact of using ensembles at various relative costs on the overall inference cost. As shown in the first plot, as we move away from the ideal, fully parallel setting with $\rho = 1$ towards the sequential setting with $\rho = 0$, the fraction of inference cost saved decreases. In fact, when the models in the cascade are of similar size—for example, when $\gamma \geq \frac{1}{5}$—a certain degree of parallelization is required for `ABC` to reduce inference cost. However, observe that when the relative costs are small enough (e.g., $\gamma \leq \frac{1}{10}$), the need for parallelization diminishes. This is evident in Figure 3 (right); for $\gamma \leq \frac{1}{50}$, the sequential execution curve ($\rho = 0$) approaches the curve with full parallelization ($\rho = 1$).

> **Takeaway #1:**
>
> - Although ensembling requires additional inference costs, these costs can be mitigated when (1) models can be parallelized or (2) smaller models are several magnitudes cheaper than larger ones. In such scenarios, `ABC` can achieve substantial accuracy improvements while reducing inference costs.

### 5.2 Real-world Use-cases

As noted in the previous two subsections, `ABC` either improves on, or is at least competitive with, the single best model in terms of accuracy. Moreover, whenever the relative cost is small enough, `ABC` suffers negligible additional penalty for using an ensemble of models. As noted in Figure 1, accuracy vs inference cost scaling for ML models already imply that the relative cost of models that only differ a few points in accuracy is small. For instance, the state-of-the-art model performance on the ImageNet-1K dataset attains about 83% top-1 accuracy with 70B FLOPs. A Pareto-optimal model that achieves 63% accuracy requires only about 1B flops, and thus a two-level `ABC` with ensembles of these two models has a relative cost of $\gamma = \frac{1}{70}$.

In many real world scenarios, this disparity in relative cost is further amplified due to deployment considerations — for instance, in an edge-to-cloud inference setting, local inter-process communication (IPC) latency is typically two orders smaller than remote (cloud) IPC latency ($\gamma \approx 10^{-2}$). Moreover, the ability of models in `ABC` to be placed on multiple, distributed devices with negligible synchronization overhead adds more opportunities for inference cost reduction. We consider three such scenarios, their respective cost models, and the benefits of `ABC` in these cases here. These include edge-to-cloud inference (§5.2.1), cloud-based model serving on heterogeneous GPUs (§5.2.2), and black-box inference access to model API services (§5.2.3).

### 5.2.1 Communication Cost in Edge-to-Cloud Inference

An advantage of `ABC` is that it allows a single large model to be split into multiple, potentially much smaller models, with only a simple reduce operation required to compute agreement. This allows us to tune device placement at various levels of `ABC` to improve inference costs. One use-case where this is beneficial is edge-to-cloud inference (Forooghifar et al., 2019); here, inference requests are generated on user-facing edge devices like mobile phones or smart devices, which are sent over the network to a cloud service for evaluation. Given an inference request generated by a user interaction, the time-to-response or response latency in such case is dominated by communication overheads (network speed, serialization overhead, network congestion, etc.) beyond our control. By using `ABC` for such applications, we are able to distribute the inference load between tiny, on-device models and the cloud models, allowing us to avoid communication costs for a significant portion of requests.

To understand the effectiveness of `ABC` in such a scenario, we consider a communication cost model previously studied in Zhu et al. (2021); Lai et al. (2022) in a setup of edge devices (i.e., Raspberry Pis and smartphones) and cloud servers. The delay parameters adopted range from small, medium, to large [1 us, 10 ms, 100 ms, 1000 ms], where near-instantaneous local communication ($< 1$ microsecond) can be expected to occur with base cascade tiers performing inference on-device, and substantial network delays might occur ($> 1$ second) in a worst-case edge-to-cloud transmission ($\gamma = 10^{-6}$).We simulate this by considering a two-level cascade, with the smaller level placed on the edge-device. We apply the delay to the cascade exit points on the edge device to capturing the time cost of transitioning between edge-to-cloud.

Our results, as shown in Figure 4, show that the flexibility that `ABC` affords in terms of model placement allows significant latency reductions, while providing superior accuracy compared to a single cloud model. In particular, we see that cascading in these scenarios provides an $14\times$ reduction in communication cost for language tasks like SST-2; and for image datasets, we see a $5\times$ reduction for ImageNet-1K and a $8\times$ reduction in CIFAR10.

> **Takeaway #2:**
> - `ABC` enables model placement flexibility where small models run locally and large models in the cloud. Communication delays can make relative costs significant, resulting in `ABC` achieving 5-14$\times$ reductions in communication costs while maintaining superior accuracy.

### 5.2.2 Monetary Cost for Model Serving on Heterogeneous Hardware

Another use-case which can take advantage of the model placement flexibility of `ABC` is using heterogeneous hardware for model serving on the cloud (Crago & Walters, 2015; Li et al., 2021a; 2023; Mo et al., 2023). GPU/Accelerator hardware typically has a disproportionately large difference in hardware costs compared to their throughput difference. For instance, based on the current pricing model offered by Lambda (Lambda, 2024), a popular cloud rental platform, the rental pricing of a single A100 is \$1.40/hour and a V100 node is \$0.06/hour ($\gamma \approx 4 \times 10^{-2}$), while the rated 32-bit tensor core throughput is 312 TFLOPs for A100 and 125 TFLOPS for V100. In this scenario, a simple placement strategy for a 2-level `ABC` that reduces inference cost may place the smaller model on V100 nodes and larger models on A100 nodes. Since the lower level models are—at least—an order of magnitude cheaper than the larger one in terms of FLOPs, this offsets the throughput loss we incur when switching from A100 to V100; and as a result, the `ABC` implementation incurs a much lower average cost.

To concretely evaluate `ABC` under such a cost model, we retrieve the costs of GPU usage by hour from Lambda Cloud's pricing (see, Table 4 in the appendix for details). For simplicity, we assume that each ensemble tier is set up on a distinct GPU in increasing order of GPU sophistication, and serves a uniform inference request rate. We also assume that the nodes are co-located and communication cost between them is negligible. We show a summary in Figure 4 and present the detailed results in Table 5, in the Appendix. We can see at least a $3\times$ reduction in inference cost in terms of \$/hour for image tasks, and a more moderate 10-30% reduction in language tasks.

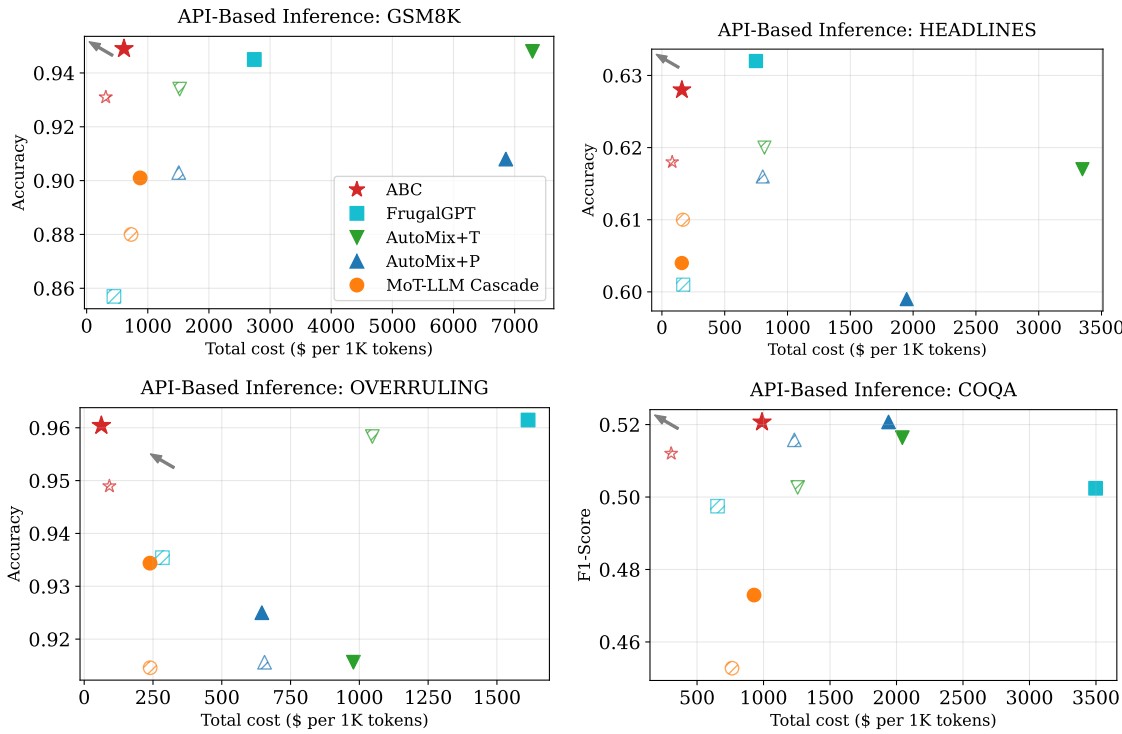

**Figure 5:** Comparison of `ABC` against state-of-the-art cascade baselines for black-box API-based inference. The faded, hatched-patterned variants represent budget-friendly, 2-level cascade instances where we do not include the costly Tier 3. Most of these methods show competitive performance, but `ABC` matches their accuracy at significantly lower costs in all tasks. Note that all these methods (aside from the MoT-LLM cascade) incur additional setup costs not reflected in our plots.

> **Takeaway #3:**
> - GPU rental cost differences significantly exceed throughput differences. `ABC` optimally places models across hardware tiers, achieving 3× cost reductions for image tasks and 10-30% savings for text tasks.

### 5.2.3 Monetary Cost for Black-box API-Based Inference

Finally, in the current landscape dominated by LLMs, many providers offer black-box API access to their proprietary LLMs (Abdalla et al., 2023; Sun et al., 2022; Hadi et al., 2024). Similarly to the large cost difference between GPU generations discussed in the previous subsection, we also observe a large cost disparity between various generations/tiers of API calls. For example, using Together.ai—one of the lowest-cost serverless endpoint providers for LLMs[1]—as a case study, we find that models within the 7B-8B range cost $0.20 per million tokens, while LlaMA3.1-405B costs $5.00. This translates to the larger model costing 25× more than the smaller model range ($\gamma = \frac{1}{25}, \rho = 1$). If we consider GPT-4 as the gold standard, the cost of usage quickly scales to 150× of our reference smaller model's costs.[2].

Since we only have black-box access to these models, we cannot employ a score-based deferral rule. However, we demonstrate that using `ABC`'s voting-based rule defined in §4 is also effective in such scenarios. For baseline comparison,

| Tier | Model | Price |
|---|---|---|
| Tier 1 | LlaMA 3.1 8B-Instruct Turbo | 0.18 |
| | Gemma 2 9B IT | 0.30 |
| | LlaMA 3 8B Instruct Lite | 0.10 |
| Tier 2 | LlaMA 3.1 70B Instruct Turbo | 0.88 |
| | Gemma 2 27B Instruct | 0.8 |
| | Qwen 2 72B-Instruct | 0.9 |
| Tier 3 | LlaMA 3.1 405B Instruct Turbo | 5.0 |

**Table 1:** The cascade tiers, their models, and associated costs, in dollar per million tokens, for the API-based experiments; all API services are provided by together.ai. We use the tiers and models as they are for `ABC` inference system. However, for the single-model cascade baselines, we use the tiers' best models for their systems.

---

[1]together.ai/pricing as of September 2024.
[2]GPT-4-1106-preview costs $30 as of September 2024.

we use FrugalGPT, 2 variants of AutoMix, and MoT LLM Cascade. These state-of-the-art cascading methods are specifically designed for scenarios where we make API calls to black-box model endpoints, in contrast to our more generally applicable `ABC` method. We access the models from Together.ai—described in Table 1—for these experiments, and consider a setting that is advantageous to the baselines, by selecting the *best singular model* from each performance tier in their respective approaches.

All the baseline methods considered here are significantly more complex and involved to set up than `ABC`; both AutoMix and FrugalGPT involve training a router or deferral rule at each cascade level which has to be repeated for every new task or model change. Unlike AutoMix, FrugalGPT requires training a DistilBERT-based scorer, which would require the user's possession of GPU resources. MoT LLM Cascade generates multiple results and varies the randomness in the LLM's responses via sampling, while using in-context demonstrations and reasoning techniques (e.g., Chain-of-Thought (Wei et al., 2022)) to influence how the model generates answers. In contrast, `ABC` uses a much simpler voting-based safe deferral rule, without involving additional training or complex routing strategies. Further method-specific details for these experiments as well as additional results can be found in Appendix D.2.

Our results, as shown in Figure 5, demonstrate that `ABC` is a more reliable deferral rule, and thus, offers a more favorable trade-off between accuracy and cost compared to existing methods—despite their more sophisticated routing mechanisms, and even without considering the additional setup costs incurred by some of the baseline methods. For instance, while FrugalGPT's scorer struggles on harder tasks and tends to take the safer route by deferring more frequently, `ABC` aggressively leverages cheaper models for a significant portion of inputs, reserving higher-cost models only when necessary. AutoMix, on the other hand, uses a few-shot self-verification system that is sampled $k$ times where $k$ is $>1$ (in the authors' codebase and ours, $k = 8$); hence, the additional API calls add significantly to its cost of usage. In contrast, `ABC` easily maintains and often improves on accuracy while being a training-free, simple approach.

> **Takeaway #4:**
> - LLM API pricing creates extreme cost ratios. `ABC`'s simple voting mechanism achieves 2-25× cost reductions compared to SOTA cascading methods without requiring complex router training.

### 5.3 Ablation Studies and Sensitivity Analysis

We conduct comprehensive ablation studies to validate `ABC`'s design choices and analyze sensitivity to key deployment parameters:

**Impact of Parallelization** §5.1.1 and Figure 8 (in Appendix E.1) analyze `ABC` under different parallelization scenarios. With full parallelization ($\rho = 1$), `ABC` achieves optimal accuracy-FLOPs trade-offs, consistently outperforming single models. Even under sequential execution ($\rho = 0$), `ABC` maintains substantial advantages when relative costs are sufficiently disparate (e.g., $\gamma \leq \frac{1}{50}$), demonstrating robustness to deployment constraints.

**Relative Cost Sensitivity** §5.1.2 and Figure 3 examine how `ABC` depends on the relative cost $\gamma$ between cascade tiers. When models have similar costs ($\gamma \geq \frac{1}{5}$), some parallelization may be required for cost savings. However, when smaller models are at least 50× cheaper ($\gamma \leq \frac{1}{50}$), `ABC` provides meaningful savings even with sequential execution, validating our theoretical analysis.

**Threshold Estimation Robustness** Appendix B and Figure 6 demonstrate that agreement threshold $\theta$ estimation is stable across different model accuracy levels (37.6% to 86.0%) and converges with minimal validation data. Using only 100 samples yields threshold estimates that remain stable when evaluated on 10× more data, confirming the effectiveness of the calibration procedure.

**Safe Deferral Rule Existence** Appendix C and Figure 7 characterize selection rates under different error tolerances (1%, 3%, 5%) across model accuracies and computational budgets. Higher-accuracy models achieve

selection rates up to 60% even with strict 1% error tolerance, empirically validating the practical existence of safe deferral rules predicted by our theory.

**Cost Breakdown Analysis**    Tables 4 & 5 (Appendix E.2) provide detailed analysis across cascade tiers, showing that most samples (52 to 93%) are processed at cheaper early tiers. This validates `ABC`'s ability to concentrate expensive computation on truly difficult samples while handling the majority of requests efficiently.

**Cascade Configuration Effects**    Figure 8 explores different cascade lengths (2-4 levels) and ensemble sizes (2-5 models per tier). The results show diminishing returns for larger ensembles, with 2-3 models per tier typically providing optimal accuracy-cost tradeoffs.

These ablations collectively demonstrate that `ABC`'s core design principles are empirically sound, with performance gracefully degrading under suboptimal deployment conditions while maintaining substantial benefits when key assumptions (e.g., large relative cost disparities) hold.

> **Takeaway #5:**
>
> - Ablation studies confirm that `ABC`'s benefits stem from the synergy of parallelization and cost disparities. Safe deferral rules exist across diverse settings, and threshold estimation requires minimal validation data ($\approx$100 samples) while remaining robust across model accuracies.

## 6    Conclusion

In this work, we introduce Agreement-Based Cascading (`ABC`) as a straightforward approach for adaptive inference that utilizes existing models for constructing cascades and makes deferral decisions based on their mutual agreement. We define safe deferral rules, ensuring `ABC` can serve as a drop-in replacement for models while improving accuracy.

Although using an ensemble of models can provide a powerful deferral rule for cascading, the additional costs required to compute such an ensemble may not lead to savings in all inference scenarios. Despite this, our work shows that this simple approach is surprisingly effective given the large differences in model sizes that reach state-of-the-art accuracy in recent ML tasks. We demonstrate improvements via a number of real-world case studies, including a study on communication costs in edge-to-cloud inference, rental costs in cloud-based settings, and the cost of black-box API services. Overall, our results demonstrate `ABC`'s capacity to improve the efficiency of adaptive inference systems without the complexities associated with traditional cascade frameworks, making it a compelling option for practitioners focused on reducing inference latency.

**Future Work.** Several promising directions emerge from this work. First, extending `ABC` to open-ended generation tasks would significantly broaden its applicability, particularly given the growing prominence of LLM-based applications. Second, exploring more efficient ensemble methods could further enhance `ABC`'s benefits in scenarios with limited parallelization capabilities, potentially through techniques that reduce ensemble overhead while maintaining agreement quality. Finally, investigating bias-aware agreement mechanisms or construction strategies that promote diversity in decision-making patterns across demographic groups could address potential bias propagation concerns inherent in majority-based voting systems.

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

## A  Agreement-Based Cascading (ABC) can Improve Accuracy

Our construction of ABC, as explained in Section 3 and 4 is based on *safe deferral rules* (Definition 4.1) — rules such that when a data point is selected at a lower cascade tier, the inference is accurate with high probability. This means that if the lower tier of the cascade has a higher accuracy compared to the largest model, on the 'easy' data it predicts on, the overall accuracy of ABC can increase. For 'hard' samples ABC and the largest, SoTA model produces identical predictions as ABC is internally deferring to this model. We formalize this notion of accuracy improvement in this section.

Consider a classification problem in the statistical learning framework, where $\mathcal{X}$ is our instance space and $\mathcal{Y}$ is the label space. Let $r$ denote any deterministic deferral rule $r : \mathcal{X} \to \{0, 1\}$. Let $\{H_1, h_2\}$ be two classifiers $h_i : \mathcal{X} \to \mathcal{Y}$ for $i \in \{1, 2\}$. We will think of $h_2$ as being the more expensive (better accuracy) model. Consider a distribution $P$ over $\mathcal{X} \times \mathcal{Y}$. The risk of classifier $h_2$ is defined as

$$\mathcal{R}(h_2) = P(h_2(x) \neq y)$$

Let us define a cascaded classifier $\mathcal{M}_r : \mathcal{X} \to \mathcal{Y}$ such that,

$$\mathcal{M}_r(x) = \begin{cases} H_1(x) & r(x) = 0 \\ h_2(x) & r(x) = 1. \end{cases} \tag{5}$$

Then the risk of the cascaded classifier using rule $r$ is,

$$\mathcal{R}(\mathcal{M}_r) = P(\mathcal{M}_r(x) \neq y)$$

We wish to understand when we can replace an existing large, expensive classifier $h_2$ with a cascade such that there is no drop in accuracy. This is useful when we are extremely sensitive towards accuracy drop and

wish to only select models at the lower-level when we are certain about the classification. Let us define the excess risk of using a cascade in-place of the larger model $h_2$ as,

$$\mathcal{R}_{\text{excess}}(\mathcal{M}_r, h_2) = \mathcal{R}(\mathcal{M}_r) - \mathcal{R}(h_2).$$

Expanding further, we can express the excess risk as,

$$
\begin{aligned}
\mathcal{R}_{\text{excess}}(\mathcal{M}_r, h_2) &= P(\mathcal{M}_r(x) \neq y) - P(h_2(x) \neq y) \\
&= P(\mathcal{M}_r(x) \neq y \mid r(x) = 0)P(r(x) = 0) + P(\mathcal{M}_r(x) \neq y \mid r(x) = 1)P(r(x) = 1) \\
&\quad - P(h_2(x) \neq y) \\
&= P(H_1(x) \neq y \mid r(x) = 0)P(r(x) = 0) + P(h_2(x) \neq y \mid r(x) = 1)P(r(x) = 1) \\
&\quad - P(h_2(x) \neq y \mid r(x) = 0)P(r(x) = 0) + P(h_2(x) \neq y \mid r(x) = 1)P(r(x) = 1) \\
&= \Big(P(H_1(x) \neq y \mid r(x) = 0) - P(h_2(x) \neq y \mid r(x) = 0)\Big)P(r(x) = 0) \quad (6)
\end{aligned}
$$

As can be seen, the general excess risk here depends on both models. Since our focus is on accuracy, any rule that leads to a risk that is no worse than that of the classifier $h_2$ can be used, and we term these as *admissible* cascades.

**Definition A.1** (Admissible cascades). *$M_r(x)$ defined above is an admissible cascade w.r.t the population distribution $P$ and the reference classifier $h_2$ if,*

$$\mathcal{R}_{excess}(M_r, h_2) \leq 0,$$

*or equivalently*

$$P(H_1(x) \neq y \mid r(x) = 0) \leq P(h_2(x) \neq y \mid r(x) = 0).$$

As mentioned in Section 4, for this work, we specialize to the case where whenever the smaller model is selected, it is always correct. That is, $P(H_1(x) \neq y \mid r(x) = 0) = 0$. Of course, such a pair of smaller-classifier $H_1$ and deferral rule $r$ need not exists. However, whenever they do, a cascade constructed using *any* $h_2$ is admissible since,

$$\forall h_2, R(M_r, h_2) = -P(h_2(x) \neq y \mid r(x) = 0)P(r(x) = 0) \leq 0.$$

This implies a *universal nature* of such $(H_1, r)$ for such two tier cascades as the excess risk of the cascade does not depend on the larger classifier $h_2$.

Finally, if the ensemble $H_1(x)$ outperforms the SoTA model on the 'easy' samples by some strictly positive $\xi > 0$, such that

$$P(H_1(x) \neq y \mid r(x) = 0) \leq P(h_2(x) \neq y \mid r(x) = 0) - \xi$$

Then, from Equation 6

$$
\begin{aligned}
\mathcal{R}_{\text{excess}}(\mathcal{M}_r, h_2) &= \Big(P(H_1(x) \neq y \mid r(x) = 0) - P(h_2(x) \neq y \mid r(x) = 0)\Big)P(r(x) = 0) \\
&= -\xi P(r(x) = 0) < 0,
\end{aligned}
$$

wherever the selection rate satisfies $P(r(x) = 0) \geq 0$. Negative excess risk implies an improvement over the SoTA classifier we were using with the overall amount of improvement depending on $P(r(x) = 0)$.

**Proof of Proposition 4.1**

**Part 1:** By Definition 4.1, we have $P(r(x) = 0, H_1^k(x) \neq y) \leq \epsilon$. The risk of the cascade decomposes as:

$$
\begin{aligned}
R(\mathcal{M}_r) &= P(r(x) = 0, H_1^k(x) \neq y) + P(r(x) = 1, h_2(x) \neq y) &(7) \\
&\leq \epsilon + P(r(x) = 1)R(h_2) \leq R(h_2) + \epsilon &(8)
\end{aligned}
$$

**Part 2:** The expected cost follows from Equation 1 and conditioning on the deferral decision:

$$E[C(\mathcal{M}_r)] = P(r(x) = 0)C(H_1^k) + P(r(x) = 1)C(h_2) = (k^\rho \gamma + P(r(x) = 1))C(h_2)$$

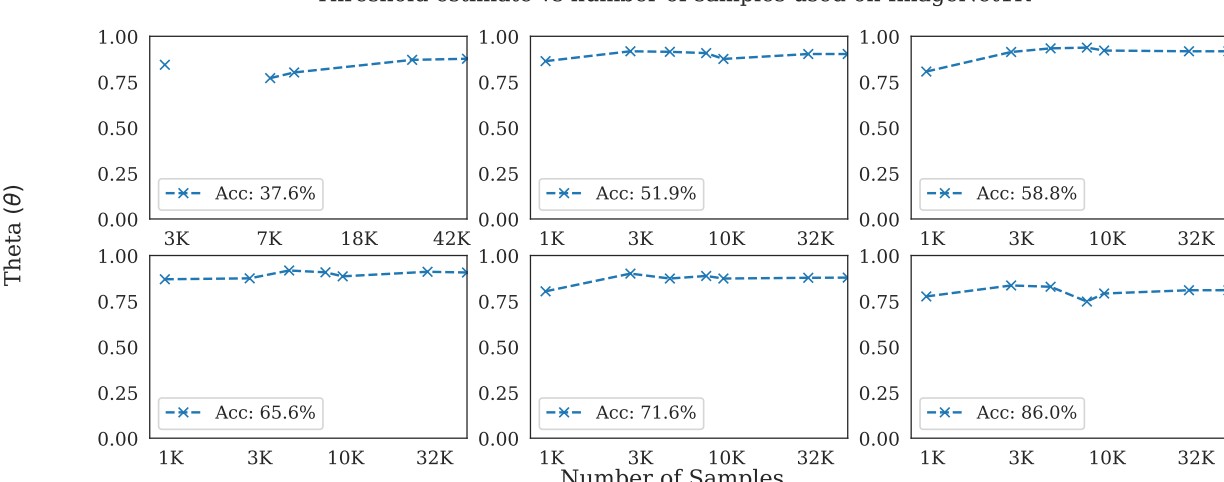

**Figure 6:** Estimation of agreement threshold $\theta$ stability as a function of the number of samples used, across different model accuracy levels on the ImageNet-1K dataset. Each plot corresponds to a model with a specific accuracy, shown in the legend. The initial estimate is with 100 samples, with subsequent estimates with larger and larger number of samples. The initial estimate is reasonably close within subsequent estimates with larger number of samples.

## B    Estimating Agreement Threshold $\theta$

To effectively apply `ABC`, it's essential to configure an agreement threshold $\theta$ for the deferral rules defined in Equations 3 and 4. This threshold indicates a sufficient level of confidence in the ensemble's predictions, allowing `ABC` to avoid deferring to a higher-cost model when the ensemble's agreement is high, thereby reducing inference costs without sacrificing accuracy. Recall from Definition 4.1 that the failure rate of a deferral rule, as a function of the agreement threshold is given by,

$$\mathbb{P}(s(x) \geq \theta, H_k(x) \neq y).$$

In particular, safe deferral rules are those with a failure rate bounded by a small $\epsilon \geq 0$ of our choice. This means that, given a distribution $\mathbb{P}$ over $\mathcal{X} \times \mathcal{Y}$ we can define a function $p(\theta)$ as,

$$p(\theta) = \mathbb{P}(s(x) \geq \theta, H_k(x) \neq y).$$

We can now define feasible thresholds, $\theta$ as those for which the error rate $p(\theta) \leq \epsilon$, since any such $\theta$ leads to safe deferral. In practice, we rarely have access to $\mathbb{P}$ and therefore $p(\theta)$. We instead use its plugin-estimator, given by

$$\hat{p}(\theta) = \frac{1}{n} \sum_{i=1}^{n} \mathbb{I}[s(x) \geq \theta, H_k(x) \neq y].$$

We estimate $\hat{p}(\theta)$ by using a small subset of samples from the validation set. Typically, we draw around 100 samples and set them aside to determine a stable threshold. Figure 6 shows how the estimated threshold varies with the number of samples used in the estimation process, across models with different accuracy levels. Each plot represents a model of a specific accuracy, with accuracy percentages indicated in the legend. As shown, the estimate generally stabilizes even with 100 samples, suggesting that only a few validation samples are needed to reliably estimate $\theta$, making the parameter estimation process efficient and practical for real-world applications.

## C    Existence of Safe Deferral Rules and Selection Rates

This section examines the existence of safe deferral rules — rules that have a very low probability of being incorrect, where data is not deferred to a larger model — by evaluating selection rates. For any threshold $\theta$,

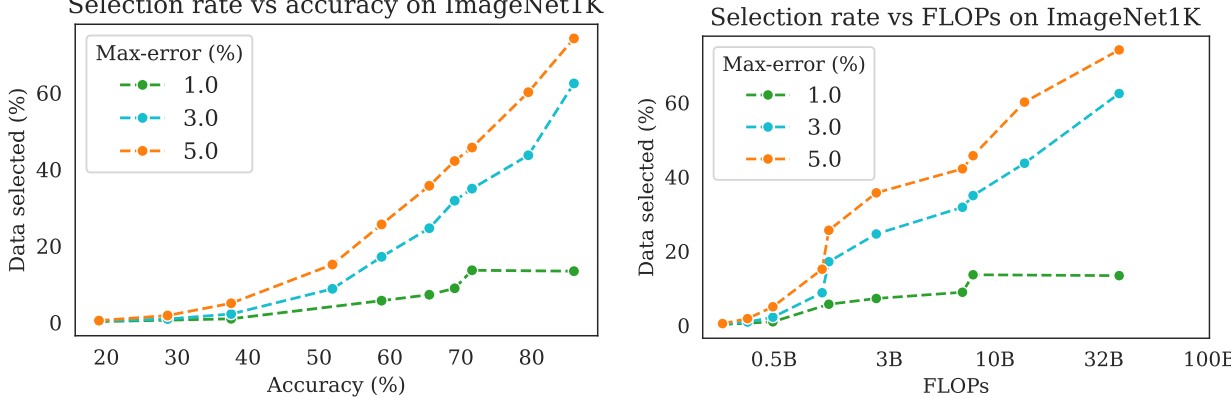

**Figure 7:** Selection rate as a function of accuracy (left) and FLOPs (right) for different error tolerances on ImageNet-1K. The selection rate $\mathbb{P}(r(x) \geq \theta)$ represents the fraction of data handled at a lower cascade tier without deferring to a larger model, based on a threshold $\theta$. Laxer error tolerances (e.g., 5%) yield higher selection rates, as more samples meet the criteria for safe deferral. In contrast, stricter tolerances (e.g., 1%) result in lower selection rates. Both plots illustrate that higher-accuracy or higher-FLOP models generally achieve higher selection rates, especially as the allowable error tolerance increases, reinforcing the stability and practicality of `ABC`.

the selection rate $P(r(x) \geq \theta)$ represents the fraction of data that deemed 'easy' enough to be processed at a lower tier of the cascade. Experimentally, we observe that $\theta_\epsilon$ computed in the manner described in Section B does indeed adhere to the error tolerance of choice $\epsilon$. This means that whenever a data point is selected at a lower tier, the inference is accurate with probability $1 - \epsilon$. Stricter values of $\epsilon$ typically result in a higher threshold for agreement and a lower-selection rate.

The plots in Figure 7 show the selection rate across different accuracy levels and FLOP values on the ImageNet-1K dataset. We explore selection rates for three error tolerances, corresponding to maximum allowable error rates of 1%, 3%, and 5%. As shown in the left plot, which compares selection rate versus accuracy of the models used in the ensemble, higher accuracy models tend to achieve higher selection rates, especially when the error tolerance is relaxed (e.g., 5% maximum error). Conversely, stricter error tolerances (e.g., 1%) lead to lower selection rates across all accuracy levels, as fewer samples meet the stringent requirements for safe deferral at lower cascade tiers. In the right plot, which illustrates selection rate versus FLOPs, a similar trend is observed. Higher-FLOP models, which are generally more accurate, are able to safely handle a larger portion of the data at lower tiers, particularly as the error tolerance increases.

# D  Evaluation Setups

## D.1  Datasets and Models

Datasets and models used in our experiments are detailed in Table 2 and Table 3.

**Table 2:** Datasets used in both benchmark and black-box API experiments across various task types.

| Category | Dataset | Task Type |
|---|---|---|
| Image Tasks | ImageNet-1K (Deng et al., 2009) | Image classification |
| | CIFAR-10 (Krizhevsky & Hinton, 2009) | Image classification |
| Language Tasks | SST-2 (Socher et al., 2013) | Sentiment analysis |
| | Twitter Financial News (Pei et al., 2021) | Sentiment analysis |
| | SWAG (Zellers et al., 2018) | Multiple-Choice QA |
| Black-Box Experiments | GSM8K (Cobbe et al., 2021) | Math Reasoning |
| | COQA (Reddy et al., 2019) | Conversational QA |
| | OVERRULING (Zheng et al., 2021) | Legal Reasoning |
| | HEADLINES (Sinha & Khandait, 2021) | News Classification |

**Table 3:** Summary of models used for both benchmark and black-box API experiments, across image and text tasks.

| Category | Dataset | Tiers Used |
|---|---|---|
| Language Models | BERT (Devlin et al., 2019)
RoBERTa (Liu et al., 2019)
XLNet (Yang et al., 2019)
ELECTRA (Radford et al., 2021) | `BASE`, `LARGE` |
| Image Models | ResNet (He et al., 2016)
ViT (Dosovitskiy, 2020)
CLIP (Clark et al., 2020) | Selection based on FLOPs |
| Black-Box Models | LlaMA 3.1 (Dubey et al., 2024)
Gemma 2 (Team et al., 2024)
Qwen 2 (Yang et al., 2024) | See Table 1 |

## D.2 Details for Black-Box API Experiments

**Baselines** We compare `ABC` to FrugalGPT (Chen et al., 2023), 2 variants of AutoMix (Madaan et al., 2023), and MoT LLM Cascade (Yue et al., 2024). Although HybridLLM (Ding et al., 2024) falls into this category of SOTA methods, it has been shown to underperform FrugalGPT and AutoMix (Madaan et al., 2023). For practical comparison, we implement all methods in a fully functional cascade system.

**Method-Specific Details**

- **AutoMix**: AutoMix trains a different router for all possible cascade steps, i.e., $n-1$ routers for the n cascade tiers, and this has to be repeated for every new task or model replacement in the system. A threshold (AUTOMIX+T) or POMDP (AUTOMIX+P) is trained with a combination of $\geq 50$ training samples from the same test data distribution and the initial inferences generated on the test data by the two models involved in each cascading step. After training the router, using the cascading system often involves running a few-shot self-verification 8 times at a high sampling temperature (temp = 1.0), using the same model that generates inference at the given cascade tier. Automix then averages the self-verification results and meta-verify it with the best parameters of the routing strategy to decide when to exit.

- **FrugalGPT**: Just like AutoMix, FrugalGPT needs to train $n-1$ routers for $n$ cascade tiers, and each router needs to have a sense of the data distribution and the model's predictive power. $\geq 500$ training samples and inference generated on these samples by the tier's model are needed to train each tier/model's scorer, a DistilBERT (Sanh et al., 2019).

- **MoT LLM Cascade**: Yue et al. (2024) focuses on sampling and consistency checking as a means of cascade. To measure consistency, the weaker LLM generates multiple answers for a single question by varying the randomness in the LLM's responses—i.e., varying the temperature of the model—while using in-context demonstrations and reasoning techniques (e.g., Chain-of-Thought (Wei et al., 2022)) to influence how the model generates answers. The system compares the different sampled answers and picks the most consistent one. If the consistency score is high enough, the weaker model's answer is accepted. Otherwise, the question is passed to the next tier.

- **ABC**: We use the voting-based safe deferral rule, requiring no additional training or any complex routing strategies.

**Cascade Models** We use the models described in Table 1. We consider a setting that is advantageous to the baselines by selecting the *best singlular model* from each performance tier and cascading between them. In all cascading systems, we use all three tiers for cascade; but considering budget constraints, we also have setups where we delete Tier 3 and use only the first two tiers (2-level cascade).

**Evaluation Setup** We evaluate these methods on a variety of (closed) generation datasets and tasks as shown in Table 2. To ensure consistency in output format and easier evaluation setup, we use few-shot

prompting—specifically, 4-shot—for all models across all tasks. For evaluation metrics, we use the macro F1 score for CoQA to capture overlaps between predictions and ground-truth answers, while we measure accuracy (essentially exact match) for the rest of the tasks. In terms of efficiency, we also measure the costs of using the model APIs. It is important to note that AutoMix and FrugalGPT incur extra setup costs that we did not factor into our results. These costs and associated latency represent a significant constraint, especially for scenarios requiring frequent retraining or adaptation to new tasks, distributions, and models.

As shown in Figure 5, we observe that `ABC` is often more cost-effective than the baselines, even with their sophisticated routing mechanisms and singular model tiers. Our analyses show that `ABC`'s efficient deferral strategy allows a more aggressive utilization of cheaper models for a significant portion of the input while only using expensive models when necessary. For instance, we realize—upon analyzing FrugalGPT's results—that the trained scorer struggles as an efficient deferral signal as the tasks get harder; hence, it is more likely to take the safer option to cascade as test sample difficulty increases. This means that `ABC` can be expected to be more efficient since the scaling laws ensure that the sum of the costs of using several cheaper models is still much less than the cost of using the larger model in the next tier. AutoMix, on the other hand, uses a few-shot self-verification system that is sampled $> 1$ times; hence, the additional API calls add significantly to its cost of usage. Considering that the self-verification process is an integral part of the AutoMix setup, it can be guaranteed that `ABC` will *always* be cheaper to use than AutoMix, despite using more models.

# E    Benefits of `ABC`

## E.1    Parallel vs. sequential inference execution

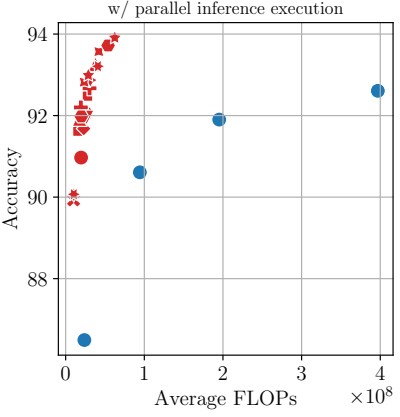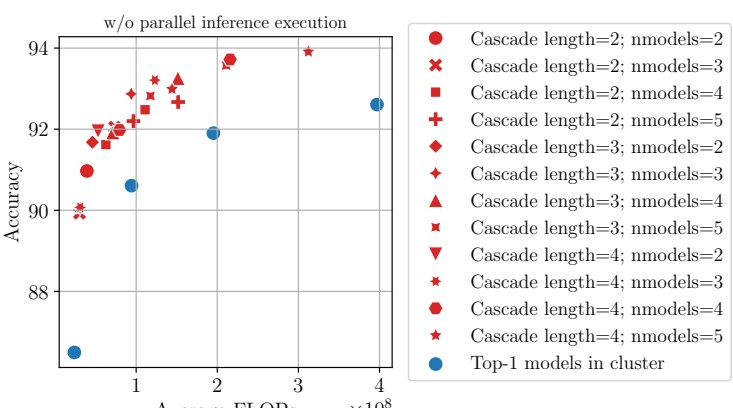

**Figure 8:** Impact of parallelization on `ABC` performance for CIFAR-10. Left: With parallel inference execution ($\rho = 1$), `ABC` configurations consistently outperform the best single models across different cascade lengths and ensemble sizes. Right: Even with sequential execution ($\rho = 0$), `ABC` maintains advantages over single models, though with reduced efficiency. The results demonstrate that while parallelization is beneficial, `ABC` remains effective even under sequential constraints when cost disparities are sufficient.

Based on Section 5.1.1, we additionally show in Figure 8 the superiority of parallel inference execution for cascading over using the best single model using CIFAR-10 as a case study. We also show that in the worst-case scenarios in which every single inference is sequentially produced over each ensemble and cascade, there are still considerable savings over the largest single models, if the scenarios assumptions are met.

## E.2    Cost Benefits

Based on Section 5.2.2, Table 4 shows the GPUs' pricing across several tiers retrieved from Lambda Cloud. Table 5 shows a detailed analysis of costs across all cascade tiers, associated with the number of cascade exits at each cascade tier, to provide holistic efficiency comparisons of the aggregated cascade costs against using the best (and largest) model — and `ABC` dominates in every measured metric. Typically, most cascade exits occur in the earlier (and much cheaper) tiers (as also shown in Table 5), ensuring that the more cost-intensive cascade tiers featuring are reserved for the harder test instances.

**Table 4:** GPU rental costs from Lambda Cloud (Lambda, 2024) (September 2024) showing the substantial cost disparities between hardware generations. The 25× cost difference between H100 and V100 GPUs, combined with more modest throughput differences, creates favorable conditions for ABC's heterogeneous hardware placement strategy described in Section 5.2.2

| GPU | Cost per Hour (USD) |
|---|---|
| V100 | 0.5 |
| A6000 | 0.8 |
| A100 | 1.29 |
| H100 | 2.49 |

**Table 5:** Detailed cost breakdown across cascade tiers for each dataset, showing the fraction of samples processed at each tier, associated GPU costs, latency, and FLOPs. The high fraction of samples processed at cheaper early tiers (52-93%) demonstrates ABC's effectiveness at concentrating expensive computation on truly difficult samples. ABC consistently outperforms single best models across all metrics while achieving substantial cost savings.

| Dataset | Metric | Tier 1 | Tier 2 | Tier 3 | Tier 4 | ABC | Best Single Model |
|---|---|---|---|---|---|---|---|
| **CIFAR-10** | Frac. Samples (total=10,000) | 0.73 | 0.09 | 0.08 | 0.10 | 1.00 | 1.00 |
| | Total GPU Cost ($ / hour) | 0.36 | 0.07 | 0.11 | 0.24 | 0.79 | 2.49 |
| | Avg. Latency (ms) | 3.11 | 3.79 | 7.76 | 9.07 | 4.13 | 9.07 |
| | Avg. FLOPs | 5.42e6 | 2.32e7 | 1.16e8 | 2.47e8 | 3.97e7 | 2.48e8 |
| **ImageNet-1K** | Frac. Samples (total=50,000) | 0.52 | 0.29 | 0.19 | - | 1.00 | 1.00 |
| | Cost ($ / hour) | 0.26 | 0.23 | 0.25 | - | 0.74 | 1.29 |
| | Avg. Latency (ms) | 2.45 | 2.88 | 3.17 | - | 2.71 | 3.17 |
| | Avg. FLOPs | 2.15e9 | 3.90e9 | 4.30e9 | - | 3.07e9 | 4.30e9 |
| **SWAG (MCQ)** | Frac. Samples (total=20,006) | 0.71 | 0.29 | - | - | 1.00 | 1.00 |
| | Cost ($ / hour) | 0.36 | 0.23 | - | - | 0.59 | 0.80 |
| | Avg. Latency (ms) | 4.52 | 8.05 | - | - | 5.53 | 8.05 |
| | Avg. FLOPs | 1.88e10 | 6.67e10 | - | - | 3.25e10 | 6.67e10 |
| **SST-2** | Frac. Samples (total=872) | 0.93 | 0.07 | - | - | 1.00 | 1.00 |
| | Cost ($ / hour) | 0.46 | 0.06 | - | - | 0.52 | 0.80 |
| | Avg. Latency (ms) | 3.88 | 7.22 | - | - | 4.13 | 7.22 |
| | Avg. FLOPs | 5.43e9 | 1.68e10 | - | - | 6.26e9 | 1.68e10 |
| **Twitter Fin News** | Frac. Samples (total=822) | 0.65 | 0.35 | - | - | 1.00 | 1.00 |
| | Cost ($ / hour) | 0.32 | 0.28 | - | - | 0.61 | 0.80 |
| | Avg. Latency (ms) | 4.05 | 7.26 | - | - | 5.19 | 7.26 |
| | Avg. FLOPs | 6.83e9 | 2.42e10 | - | - | 1.30e10 | 2.42e10 |

