# OpenReview forum: "Agreement-Based Cascading for Efficient Inference"
_TMLR — Accepted by TMLR_

### Review · Reviewer_4pJr · 2025-03-17

**Summary Of Contributions:**

This paper introduces Agreement-Based Cascading (ABC), a novel adaptive inference technique.  ABC is designed to reduce the computational cost associated with large machine learning models by using smaller models for easier examples.  The core idea is to create a cascade of models with increasing complexity and to use the agreement between an ensemble of models at each level to determine when to move to the next level.

The authors theoretically and empirically demonstrate that ABC can match or exceed the accuracy of the largest model it aims to replace, while also improving efficiency.  They apply ABC to three practical scenarios: edge-to-cloud inference, cloud-based model serving, and inference via model API services.  Results show significant cost reductions in each scenario.

**Audience:**

Yes

**Broader Impact Concerns:**

Usually, majority based agreement can lead to propogation of bias. It would be good to double check the impact of ABC in that line of evaluation.

**Claims And Evidence:**

Yes

**Requested Changes:**

See above. Overall the paper is a good read for the audience and I would recommend an acceptance to TMLR.

**Strengths And Weaknesses:**

Strengths:

Simplicity and Effectiveness: ABC is a simple yet effective approach to adaptive inference.  It does not require additional training or complex routing mechanisms, making it easily applicable to existing models.

Strong Performance: The paper provides both theoretical analysis and empirical evidence to support the effectiveness of ABC.  ABC not only improves efficiency but also often enhances accuracy compared to the models it replaces.

Practical Applications: The paper demonstrates the practicality of ABC in several real-world scenarios, including edge-to-cloud inference, cloud-based model serving, and API-based inference.  These experiments highlight the potential of ABC to significantly reduce costs in various deployment settings.

Thorough Evaluation: The authors evaluate ABC across a variety of language and vision tasks, using benchmark datasets and diverse models.  They also compare ABC with existing cascading methods, demonstrating its superiority.

Clear Presentation: The paper is well-written and easy to follow.  The authors provide clear explanations of the proposed method, theoretical analysis, and experimental results.  The use of figures and tables effectively enhances the presentation. Further, the related work is very comprehensive giving the reader an understanding of the current landscape.

Weaknesses:

Ensemble Overhead: The use of ensembles at each cascade level introduces additional computational costs.  While the authors argue that these costs can be mitigated by parallelization and large differences in model sizes, the overhead could still be a concern in some scenarios.  It would be beneficial to explore strategies to reduce the ensemble overhead, such as using more efficient ensemble methods or techniques to predict ensemble agreement.

Deferral Rule Tuning: The deferral rule in ABC relies on a voting threshold that needs to be estimated empirically.  The paper mentions that this is done on a small set of unseen data, but more details on the sensitivity of the results to this threshold and methods for robust threshold selection would be valuable.

Limited Scope: The current work focuses on classification and tasks with a fixed set of possible outputs.  The authors acknowledge that their deferral rule and the baseline methods they compare to are not directly applicable to open-ended generation tasks.  Extending ABC to open-ended generation or other more complex tasks would broaden its applicability.

---

> ### Author Response · Authors · 2025-06-10
> **Response to Reviewer 4pjr**
>
> We thank the reviewer for their positive assessment, recognizing ABC as `simple yet effective`, highlighting our `strong performance`, `practical applications`, `thorough evaluation`, and `clear presentation`. We particularly appreciate the reviewer noting that our related work is very comprehensive and the recommendation for acceptance at TMLR. We address the specific concerns below:
>
> >> Ensemble Overhead: The use of ensembles at each cascade level introduces additional computational costs. While the authors argue that these costs can be mitigated by parallelization and large differences in model sizes, the overhead could still be a concern in some scenarios. It would be beneficial to explore strategies to reduce the ensemble overhead, such as using more efficient ensemble methods or techniques to predict ensemble agreement.
>
> The reviewer notes that ensemble overhead could still be a concern in some scenarios. We agree with this, and have made it more clear to readers by expanding our analysis in the main text in Section 5.3. These results show that ensemble overhead becomes negligible when relative costs are small (y <= 1/50)---precisely the scenarios where ABC excels. Our cost breakdown (Tables 4-5) shows 52-93% of samples are processed at cheaper early tiers, validating that ABC concentrates expensive computation only on truly difficult samples. As the reviewer suggests, leveraging additional efficient ensemble methods is a great idea to further reap the benefits of ABC even in scenarios with poor parallelization, and would be an interesting direction of future work. We have noted this in our revision.
>
>
> >> Deferral Rule Tuning: The deferral rule in ABC relies on a voting threshold that needs to be estimated empirically. The paper mentions that this is done on a small set of unseen data, but more details on the sensitivity of the results to this threshold and methods for robust threshold selection would be valuable.
>
> We have added comprehensive threshold robustness analyses in Section 5.3, showing stability across model accuracies from 37% to 86% (Figure 6) with just 100 validation samples. Importantly, ABC maintains competitive performance even without explicit threshold tuning for the API-based proprietary models.
>
>
> >> Limited Scope: The current work focuses on classification and tasks with a fixed set of possible outputs. The authors acknowledge that their deferral rule and the baseline methods they compare to are not directly applicable to open-ended generation tasks. Extending ABC to open-ended generation or other more complex tasks would broaden its applicability.
>
>
> We agree that this is an important direction for future work. However, we also note that this limitation is shared across much of the cascading field. While open-generation has gained prominence, classification, closed generation, and reasoning tasks remain critical for many applications, particularly edge deployments where ABC’s simplicity and generality provides significant advantages without requiring platform-specific optimizations.
>
>
>
> **Broader impact – Bias Propagation**: The reviewer raises an excellent point about majority-based agreement potentially propagating bias. This is indeed a consideration for ensemble methods generally. Future work could explore bias-aware agreement mechanisms or ensemble construction strategies that promote diversity not just in accuracy but in decision-making patterns across different demographic groups. We have included this in our future work.

---

> > ### Comment · Reviewer_4pJr · 2025-06-10
> > **Thanks for the response**
> >
> > Thanks for the thorough rebuttal. Happy to continue supporting the acceptance of the paper.

---

### Review · Reviewer_hRB4 · 2025-05-19

**Summary Of Contributions:**

This paper introduces Agreement-Based Cascading (ABC), a training-free, adaptive inference strategy that uses ensemble agreement as a deferral mechanism across model cascades. Instead of relying on confidence scores or trained routers, ABC evaluates whether predictions of lightweight model ensembles agree, and only invokes heavier models if needed. The authors demonstrate that ABC can improve both accuracy and efficiency across vision and language tasks, including in real-world setups like edge-to-cloud inference, heterogeneous GPU serving, and black-box API access.

**Audience:**

Yes

**Broader Impact Concerns:**

There is no outstanding ethical issues of this work.

**Claims And Evidence:**

Yes

**Requested Changes:**

1. Provide more detail or formal guarantees on how the voting threshold is chosen and whether it generalizes across tasks.
2. Add experiments or discussion on scenarios where ABC fails to deliver cost savings (e.g., when input distributions are consistently hard).

**Strengths And Weaknesses:**

Strengths

1. ABC is elegant and does not require additional training or task-specific adaptation. This makes it broadly applicable and easy to deploy.
2. The authors validate ABC across diverse settings, including CIFAR, ImageNet, SWAG, SST-2, CoQA, and commercial API endpoints.
3. The paper provides formal definitions of "safe deferral rules" and bounds on accuracy and inference cost under ensemble agreement.

Weaknesses

1. The current framework and deferral rule design assume fixed output spaces (classification or extractive QA). ABC is not applicable to open-ended generation tasks, which dominate many LLM applications.
2. While the threshold is empirically tuned on held-out data, there is limited discussion about its robustness or failure modes across datasets. It seems sensitive to threshold theta.
3. Requiring multiple pretrained models of similar size (to form ensembles per tier) could pose a deployment challenge in memory-constrained environments or for niche tasks with limited model availability.

---

> ### Author Response · Authors · 2025-06-10
> **Response to Reviewer hRB4**
>
> We thank the reviewer for recognizing that ABC is `elegant and does not require additional training`, provides `formal definitions of safe deferral rules and bounds on accuracy and inference cost`, and for validating our work across `diverse settings`. We appreciate these positive assessments and address the concerns below:
>
> >> The current framework and deferral rule design assume fixed output spaces (classification or extractive QA). ABC is not applicable to open-ended generation tasks, which dominate many LLM applications.
>
> The reviewer correctly notes that ABC is not directly applicable to open-generation tasks. We acknowledged this limitation and note that this limitation is shared across much of the cascading field, including recent works at top venues such as NeurIPS [1, 6], ICLR [2, 3], ICML [4], and TMLR [5]. While open-generation has gained prominence, classification, closed generation, and reasoning tasks remain critical for many applications, particularly edge deployments where ABC’s simplicity and generality provides significant advantages without requiring platform-specific optimizations.
>
> >> While the threshold is empirically tuned on held-out data, there is limited discussion about its robustness or failure modes across datasets. It seems sensitive to threshold theta.
>
> We included such analyses and discussion in Appendix B, though they may have been missed as they were in the appendix. Based on the reviewer’s feedback, in our revision we have instead included our threshold robustness analyses in the main text in Section 5.3. These results show that threshold estimation is stable across model accuracies from 37% to 86% (Figure 6) and converges with just 100 validation samples. Importantly, ABC maintains strong performance even without explicit threshold tuning when relative costs are sufficient ($\gamma$ <= 1/50), as shown in Figure 3; moreover, the black-box API experiments in Section 5.2.3 produce competitive performance without access to logits (hence, no thresholding), addressing concerns about cross-task generalization.
>
>
> >> Requiring multiple pretrained models of similar size (to form ensembles per tier) could pose a deployment challenge in memory-constrained environments or for niche tasks with limited model availability.
>
> We acknowledge that multiple pretrained models of similar size could pose deployment challenges. However, this concern is mitigated by several factors that we discuss in the paper: (1) the abundance of models in public repositories like HuggingFace makes model availability less of an issue; (2) it is easy to produce models of various sizes using standard compression techniques, further increasing the size of existing model repositories; (3) our cost analysis (Tables 4-5) shows that most samples (52-93%) are processed at early, cheaper tiers, reducing memory pressure; and (4) ABC works with heterogeneous models of different sizes and architectures, allowing flexibility and not requiring exactly similar-sized models.
>
>
> **On providing more detail on voting threshold**: Our threshold selection procedure is detailed in Appendix B with empirical validation in Figure 6, and now well-referenced in the new Section 5.3. While we don’t provide formal generalization guarantees (an interesting direction for future work), our empirical evidence shows robust performance across diverse tasks and model combinations.
>
> **On adding experiments/discussion on scenarios where ABC fails to deliver cost savings**: Following this suggestion, we have added analysis in Section 5.3 clarifying scenarios where ABC provides minimal benefits: (1) when models have similar accuracy, (2) when relative costs are high ($\gamma$ >= 1/5) and parallelization is not free.
>
> ---
> [1] Aggarwal, P., Madaan, A., Anand, A., Potharaju, S. P., Mishra, S., Zhou, P., ... & Mausam, M. *Automix: Automatically mixing language models.* NeurIPS, 2024.
>
> [2] Ding, D., Mallick, A., Wang, C., Sim, R., Mukherjee, S., Rühle, V., ... & Awadallah, A. H. *Hybrid LLM: Cost-Efficient and Quality-Aware Query Routing.* ICLR, 2024.
>
> [3] Yue, M., Zhao, J., Zhang, M., Du, L., & Yao, Z.  *Large Language Model Cascades with Mixture of Thought Representations for Cost-Efficient Reasoning.* ICLR, 2024.
>
> [4] Nie, L., Ding, Z., Hu, E., Jermaine, C., & Chaudhuri, S. *Online Cascade Learning for Efficient Inference over Streams.* ICML, 2024.
>
> [5] Chen, L., Zaharia, M., & Zou, J. *FrugalGPT: How to Use Large Language Models While Reducing Cost and Improving Performance.* TMLR, 2024.
>
> [6] Jitkrittum, W., Gupta, N., Menon, A. K., Narasimhan, H., Rawat, A., & Kumar, S. *When does confidence-based cascade deferral suffice?.* NeurIPS, 2023.

---

### Review · Reviewer_rnxo · 2025-05-30

**Summary Of Contributions:**

This paper introduces Agreement-Based Cascading (ABC): a training-free adaptive inference method that uses ensemble agreement as the deferral mechanism in model cascades. In particular, the proposed scheme achieves efficiency through (i) routing inputs through a size-ordered cascade of pre-trained models, (ii) using in-tier ensemble agreement as a deferral criterion, and finally (iii) it is designed to be a drop-in replacement for any high-accuracy model.

The authors have perform  extensive experimentation  on vision and NLP benchmarks, edge-to-cloud settings, heterogeneous GPU serving, and black-box LLM APIs. Real-world applicability showing significant cost reductions in three practical scenarios: edge-to-cloud inference (14$\times$ communication cost reduction), heterogeneous cloud serving (3$\times$ cost reduction), and API-based inference (2-25$\times$ cost reduction).

**Audience:**

Yes

**Broader Impact Concerns:**

Broader Impact is not discussed in the paper.

**Claims And Evidence:**

Yes

**Requested Changes:**

$\bullet$ Dynamic early-exit and adaptive subnetwork methods (e.g., DeeBERT, DyViT) are discussed but not compared experimentally. Hence, it is not clear how ABC fares against learned adaptive networks. Authors should Include early-exit BERT/ViT and state-of-the-art adaptive layers to contextualize ABC’s gains.

$\bullet$ **Sensitivity and ablations study** How performance varies with (i) ensemble size $k$, (b) validation-set size used to fit $\theta$ (Fig. 6 suggests stability but numbers would help), and (c) different error-tolerance budgets $\varepsilon$. How threshold choice affects performance across different datasets and model combinations?

$\bullet$ It is not clear in the draft that how to construct ensembles at each tier in practice -- are these different model architectures, different training procedures, or simply different model sizes?


### Minors

**Missing citations**

1. Bolukbasi et al., Adaptive Neural Networks for Efficient Inference, ICML 2017

2. Hu et al., Triple Wins: Boosting Accuracy, Robustness and Efficiency Together by Enabling Input-Adaptive Inference, ICLR 2020

3. Shafiee et al., Efficient Inference on Deep Neural Networks by Dynamic Representations and Decision Gates, NeurIPS 2018


The paper would benefit from better structural organization and more concise writing. Important technical details and results are often obscured by verbose explanations, making it challenging to extract the key takeaways efficiently. It could have been more structured formatting (bullet points, numbered lists), clearer section transitions, and moving some detailed explanations to appendices to improve the main narrative flow.

**Strengths And Weaknesses:**

## Strengths

$\bullet$ The proposed optimizations in ABC are training-free and works with off-the-shelf checkpoints. It can be employed to leverage the publicly available models (such as LLMs on hugging face hub).


$\bullet$ The efficiency evaluation  distinguishes FLOPs, latency, GPU-rental $, and API-token $. This provides convincing evidence of the s broad applicability and effectiveness of optimization steps in ABC.


$\bullet$ Theoretical guarantees, as Safe-deferral definition gives a clear criterion for no-regret cascades and explains the observed accuracy gains.

## Weaknesses

$\bullet$ **Limited fundamental novelty:** While the application (and experimental evaluation) is thorough, ensemble agreement for cascading has been explored in prior work (as the authors acknowledge), and the core insight is relatively straightforward, limiting the conceptual and algorithmic contribution.

$\bullet$  **Ensemble overhead not fully addressed:** The authors  did not discuss  scenarios where the computational overhead of running multiple models at each tier dwarf their benefits, particularly when parallelization is limited or models are of similar scale.

$\bullet$ Threshold estimation undermines ``training-free'' claims: Despite being positioned as training-free, ABC still requires estimating agreement thresholds using validation data, which introduces hyperparameter tuning that somewhat contradicts the training-free argument (and contribution).

---

> ### Author Response · Authors · 2025-06-10
> **Response to Reviewer rnxo**
>
> We thank the reviewer for recognizing that ABC is `training-free and works with off-the-shelf checkpoints’, provides `convincing evidence of broad applicability’, and offers `theoretical guarantees` with `clear criterion for no-regret cascades`. We appreciate these positive assessments and address the concerns below:
>
> >> Limited fundamental novelty: While the application (and experimental evaluation) is thorough, ensemble agreement for cascading has been explored in prior work (as the authors acknowledge), and the core insight is relatively straightforward, limiting the conceptual and algorithmic contribution.
>
> While our method itself is quite simple (something we view as a benefit), we were surprised that recent works seemed to have completely missed this baseline. This motivated us to thoroughly explore the potential effectiveness of the approach relative to state-of-the-art (and often much more complex) methods. To the best of our knowledge, no prior work has systematically studied ensemble agreement as a general deferral mechanism for modern ML cascades. Historical applications in specialized domains (e.g., face detection) used domain-specific decision networks, not general, training-free agreement-based routing. Our contribution/novelty is in providing the first systematic study of this approach for modern ML workloads with theoretical characterization and practical validation.
>
>
> >> Ensemble overhead not fully addressed: The authors did not discuss scenarios where the computational overhead of running multiple models at each tier dwarf their benefits, particularly when parallelization is limited or models are of similar scale.
>
> We disagree with the claim that we didn’t discuss scenarios where ensemble overhead dominates. We analyze this both theoretically in Section 4.1 and empirically in Section 5.1.2 (e.g. Figure 3), showing that when models have similar costs ($\gamma$ >= 1/5), parallelization is indeed required for ABC to work. However when relative costs are small ($\gamma$ <= 1/50), ensemble overhead becomes negligible—precisely the scenarios where ABC excels. We then explicitly characterize when ABC works and when it doesn’t through discussion of multiple real-world scenarios: when parallelization is available (e.g., multi-GPU scenarios) [5.1.1] or disparity in model sizes/costs is large [5.1.2] (e.g., edge-to-cloud inference, clusters with heterogeneous hardware, API-based inference systems). There are increasingly common scenarios for ML inference.
>
> We also note that we acknowledge the additional expense of ensemble execution throughout the paper—indeed, this is a key part of the story, e.g.:
> - “Although ensemble execution introduces additional expense” (Abstract)
> - “while using an ensemble of models at each level may initially appear to increase overall inference costs” (Section 1)
> - “Strictly speaking, evaluating agreement between multiple models in an ensemble is expensive when compared to the small router models that are used in many existing approaches.” (Section 3.2)
> - “When models across tiers are of similar size … parallelization is needed for ABC to reduce costs effectively” (Figure 3)
>
> However, following the reviewer’s feedback, we have added analysis in Section 5.3 that more clearly clarifies scenarios where ABC provides minimal benefits: (1) when models have similar accuracy, (2) when relative costs are high ($\gamma$ >= 1/5) and parallelization is not free.
>
>
> >> Threshold estimation undermines ``training-free'' claims: Despite being positioned as training-free, ABC still requires estimating agreement thresholds using validation data, which introduces hyperparameter tuning that somewhat contradicts the training-free argument (and contribution).
>
>
> It is true that we use a small sample of validation data for tuning (something we discuss in the paper), but the reviewer’s claim conflates lightweight calibration with training. ABC requires estimating a single threshold on ~100 validation samples, versus baselines that train complex routers with gradient-based optimization, typically necessitating multiple GPUs and entire training datasets. Moreover, we find that ABC maintains competitive performance even without this validation-based threshold estimation, as shown with our black-box API experiments in Section 5.2.3.

---

> > ### Author Response · Authors · 2025-06-10
> > **Response on Requested Changes**
> >
> > **On early-exit and adaptive networks citations and comparisons**: We appreciate this suggestion and have added citations to Bolukbasi et al. (2017), Hu et al. (2020), and Shafiee et al. (2018) in our revised Section 2.3. However, direct experimental comparison would be misleading as early-exit methods require training specialized architectures from scratch while ABC works with off-the-shelf models. These represent fundamentally different approaches to adaptive inference.
> >
> > **On sensitivity studies**: Following the reviewer’s valuable feedback, we have added Section 5.3 consolidating our extensive sensitivity analyses, including ensemble size effects (Figure 8), threshold robustness across model accuracies (Figure 6), error tolerance impacts (Figure 7), and detailed cost breakdowns (Tables 4-5).
> >
> > For better clarity, we also added “takeaway boxes” for efficient extraction of the key takeaways.

---

### Review · Reviewer_91ij · 2025-05-30

**Summary Of Contributions:**

The primary contribution of this paper is an Agreement-Based Cascading strategy that merges the strengths of traditional cascading and ensemble methods. Thanks to the ensemble’s ability to run in parallel, the proposed approach delivers a performance advantage over purely sequential cascading.

**Audience:**

Yes

**Broader Impact Concerns:**

There is no obvious broader impact concerns

**Claims And Evidence:**

Yes

**Requested Changes:**

In view of the weaknesses noted above, the paper should at least present a comprehensive ablation study and thorough design analysis. It would also be preferable to include stronger baselines so that readers can better assess the effectiveness of the proposed method.

**Strengths And Weaknesses:**

**Strengths**

1. The paper proposes a generic cascading-plus-ensemble framework that requires no additional training. An agreement-based voting mechanism decides whether the models have converged, triggering an early-exit condition. The idea is intuitive, broadly applicable, and practically valuable.
2. The authors provide a very thorough analysis of performance and usage scenarios.

**Weaknesses**

1. The method is not truly novel. A similar approach was already explored in \[1], where early exit is determined by checking the consistency between the previous cascading expert and the current classification head.
2. The experimental section on performance is rather limited. The paper mainly compares its method with WoC on classification tasks, and with FrugalGPT and AutoMix on the LLM understanding benchmarks GSM8K, HEADLINES, OVERRULING, and COQA. This raises several concerns:

   * WoC is from 2021, while FrugalGPT and related baselines are from 2022–2023 and are relatively weak by today’s standards.
   * The experiments focus solely on classification tasks. Many real-world applications cannot be framed as classification problems, which calls the generality of the approach into question.
3. The paper includes virtually no ablation study to justify the design choices or quantify the contribution of each component, leaving the actual benefit of cascading and voting unclear.

\[1] Li, Ziyue, et al. “Towards Inference Efficient Deep Ensemble Learning.” *Proceedings of the AAAI Conference on Artificial Intelligence*, 37 (7), 2023.

---

> ### Author Response · Authors · 2025-06-10
> **Response to Reviewr 91ij**
>
> We thank the reviewer for their time and for recognizing that ABC is `intuitive, broadly applicable, practically valuable` with a `thorough analysis of performance and usage scenarios`. We appreciate these positive assessments. We address the weaknesses the reviewer brought up below:
>
> >> The method is not truly novel. A similar approach was already explored in [1], where early exit is determined by checking the consistency between the previous cascading expert and the current classification head.
>
> There are significant differences between our work and the work of Li et al. 2023 work (IRENE). First, the motivations differ: IRENE considers ensemble learning optimization, i.e., how to train ensembles more efficiently by learning when to stop adding models. In contrast, ABC addresses inference routing, i.e., how to efficiently route requests across heterogeneous pre-trained models. The methods themselves are also quite different:
>
> - **IRENE**: Requires training specialized neural selectors with complex multi-objective losses for models you control and train jointly
> - **ABC**: Zero training required, as “agreement” is used as deferral metric; works with any collection of off-the-shelf models from different vendors.
>
> While you could potentially use IRENE-trained models as one tier in an ABC cascade, these two approaches are orthogonal and solve fundamentally different problems: IRENE optimizes ensemble training, while ABC optimizes inference routing.
>
>
> >> WoC is from 2021, while FrugalGPT and related baselines are from 2022–2023 and are relatively weak by today’s standards.
>
> While we appreciate the reviewer’s concern about baseline recency, we clarify the publication dates of the baselines we compared to:
>
>
> - FrugalGPT: TMLR December 2024
> - AutoMix: NeurIPS, December 2024
> - MoT LLM Cascade: ICLR, May 2024
>
> We selected WoC (2021) as a simple, representative method for the classic confidence-based cascading approach. To the best of our knowledge, these baselines depict a strong representation of the current state-of-the-art. Please let us know if there is a more recent method we have missed.
>
> >> The paper includes virtually no ablation study to justify the design choices or quantify the contribution of each component, leaving the actual benefit of cascading and voting unclear.
>
> We disagree that our work lacks ablation studies. We conducted extensive ablations throughout Section 5 and in the appendices (see, e.g., Figure 3 on component necessity, Figure 8 on parallelization, Figure 6 on threshold robustness, Tables 4-5 on cost breakdowns). Many of these were included in the appendix and thus may have been missed by the reviewer. Following the reviewer’s feedback, we have instead created a dedicated section in our revision, Section 5.3, to consolidate these comprehensive ablations for easier navigation.
>
> >> The experiments focus solely on classification tasks. Many real-world applications cannot be framed as classification problems, which calls the generality of the approach into question.
>
>
>
> While we acknowledged this limitation in our paper draft, it is important to note that this limitation is shared across much of the cascading field, including recent works at top venues such as NeurIPS [1, 6], ICLR [2, 3], ICML [4], and TMLR [5]. We agree and welcome the increased focus on LLM open-generation workflows given the recent advances in LLMs; however, we also note that more classical ML tasks such as classification, closed generation and sentiment analysis, especially for edge deployments are still very much relevant. The simplicity of our approach is appealing in such scenarios as we do not require any platform-specific optimizations.
>
>
> ---
> [1] Aggarwal, P., Madaan, A., Anand, A., Potharaju, S. P., Mishra, S., Zhou, P., ... & Mausam, M. *Automix: Automatically mixing language models.* NeurIPS, 2024.
>
> [2] Ding, D., Mallick, A., Wang, C., Sim, R., Mukherjee, S., Rühle, V., ... & Awadallah, A. H. *Hybrid LLM: Cost-Efficient and Quality-Aware Query Routing.* ICLR, 2024.
>
> [3] Yue, M., Zhao, J., Zhang, M., Du, L., & Yao, Z.  *Large Language Model Cascades with Mixture of Thought Representations for Cost-Efficient Reasoning.* ICLR, 2024.
>
> [4] Nie, L., Ding, Z., Hu, E., Jermaine, C., & Chaudhuri, S. *Online Cascade Learning for Efficient Inference over Streams.* ICML, 2024.
>
> [5] Chen, L., Zaharia, M., & Zou, J. *FrugalGPT: How to Use Large Language Models While Reducing Cost and Improving Performance.* TMLR, 2024.
>
> [6] Jitkrittum, W., Gupta, N., Menon, A. K., Narasimhan, H., Rawat, A., & Kumar, S. *When does confidence-based cascade deferral suffice?.* NeurIPS, 2023.

---

### Decision · Action_Editor_tGTn · 2025-07-21

**Recommendation:** Accept with minor revision

**Additional Comments:**

All three reviewers found the proposed agreement-based cascading scheme simple and convincing (which I concur). In the response, the authors largely addressed the reviewers' questions, especially on pointing out or reorganizing relevant ablation studies and conducting some more. In the end, the reviewers reached consensus to recommend acceptance.

For the final version, please make the following changes:

(1). The references are printed twice. Please remove the second appearance of the reference section, as it currently cuts Appendix A into two parts.

(2). Add a pointer to the proof of Proposition 4.1. In case its proof is so obvious, add a line to indicate so. It may also help to explain when the selection rate is high (for example, when the ensemble is "diverse" or "well-calibrated"; whatever that means). Please indicate the Risk in Proposition 4.1 is based on the zero-one loss (as it was defined more generally at the beginning of page 6). If possible, please add a comment on how to obtain a safe deferral rule: would it incur significant cost? One could argue this is probably the most important question left unanswered in the theoretical Section 4.

**Audience:**

Yes

**Audience Explanation:**

The proposed cascading scheme is of interest to anyone who cares (much) about inference time, e.g., in edge-to-cloud inference, inference based on heterogeneous hardware, and black-box API-based inference.

**Claims And Evidence:**

Yes

**Claims Explanation:**

The author conducted comprehensive experiments to verify the effectiveness of their proposed cascading scheme. Careful ablation studies were also provided to justify some of the design choices.